# Cortical neural dynamics unveil the rhythm of natural visual behavior in marmosets

Takaaki Kaneko [1,2,6✉], Misako Komatsu [3,6], Tetsuo Yamamori[3], Noritaka Ichinohe[4] & Hideyuki Okano [1,5✉]

Numerous studies have shown that the visual system consists of functionally distinct ventral and dorsal streams; however, its exact spatial-temporal dynamics during natural visual behavior remain to be investigated. Here, we report cerebral neural dynamics during active visual exploration recorded by an electrocorticographic array covering the entire lateral surface of the marmoset cortex. We found that the dorsal stream was activated before the primary visual cortex with saccades and followed by the alteration of suppression and activation signals along the ventral stream. Similarly, the signal that propagated from the dorsal to ventral visual areas was accompanied by a travelling wave of low frequency oscillations. Such signal dynamics occurred at an average of 220 ms after saccades, which corresponded to the timing when whole-brain activation returned to background levels. We also demonstrated that saccades could occur at any point of signal flow, indicating the parallel computation of motor commands. Overall, this study reveals the neural dynamics of active vision, which are efficiently linked to the natural rhythms of visual exploration.

[1] Laboratory for Marmoset Neural Architecture, RIKEN Center for Brain Science, Saitama, Japan. [2] Systems Neuroscience Section, Primate Research Institute, Kyoto University, Aichi, Japan. [3] Laboratory for Molecular Analysis of Higher Brain Function, Center for Brain Science, RIKEN, Saitama, Japan. [4] Department of Ultrastructural Research, National Institute of Neuroscience, National Center of Neurology and Psychiatry, Tokyo, Japan. [5] Department of Physiology, Keio University School of Medicine, Tokyo, Japan. [6]These authors contributed equally: Takaaki Kaneko, Misako Komatsu. ✉email: kaneko.takaaki.6w@kyoto-u.ac.jp; hidokano@a2.keio.jp

The primate visual system is one of the most investigated cortical circuitries. More than 30 cortical areas have visual functions and are organized hierarchically into complex feedforward and feedback connections[1,2]. The dual-stream hypothesis models how visual information is processed in this circuitry, in which the visual input received by the retina is transmitted to the primary visual cortex (V1) and then flows to the functionally distinct dorsal and ventral visual streams[3-5]; the former is dedicated to the analysis of scenes and object semantics[6], while the latter is important for the analysis of the spatial properties of visual information[7].

However, several recent studies have shed light on a complementary or alternative schema for visual information flow proposed in this theory. First, functional[8-10] and anatomical studies[11-14] have shown that there are many alternative routes that bypass V1, which is often assumed to be the entry point for visual information into the cerebral cortex, such as the extrastriate cortex and temporoparietal regions. Second, recent advances in our understanding of occipital and temporoparietal white matter revealed that communication across the dorsal and ventral visual areas is far richer than previously thought[15,16]. Third, natural visual behavior in primates is an intrinsically recurrent process consisting of recursive sampling of the same visual scene with different eye positions, and saccades are the result of ongoing visual processing. This forms a circular process of visual computation, a cognitive decision for the next saccade target, and motor execution[17,18]. While the neural mechanisms underlying each step of this behavioral sequence have been studied in detail, the neural dynamics of the entire cycle of active vision have been described rarely. A precise description of the timing at which cortical areas are activated and which regions exchange signals under active vision is essential for the construction of a computation model for active vision.

To capture the neural dynamics in the cerebral cortex during natural visual behavior, it is crucial to examine the exact spatio-temporal dynamics and trajectories of neural signal around saccades and the interactions of multiple areas during active visual exploration. To address these issues, we recorded the cortical neural dynamics of natural visual behavior in marmoset monkeys by using an electrocorticographic (ECoG) array covering almost the entire lateral hemisphere, over 63 cortical areas with 96 electrodes at an inter-electrode distance of 2.5 mm[19], while marmosets freely viewed naturalistic movie stimuli (Fig. 1a, b). Marmosets are an ideal primate model for this type of study as their brain shares most of the organizational features of the visual system found in other primates[15,20-23], and their smooth brain surface without complex sulci allows cortical-wide recording using ECoG electrodes. In this study, we found that the dorsal stream was activated prior to the ventral stream under natural visual behavior. The information traversed the cortical sheet from the dorsal to ventral areas. Furthermore, we identified the recursive nature of active vision in which neural architecture and visual behavior are coordinated for the efficient exploration of visual scenes.

## Results

**Signal flow under natural visual behavior.** During natural visual behavior, the marmosets performed rapid eye movements (saccades) to sample a visual scene with different eye positions every 220 ms (Fig. 1c, d). These saccades refresh the visual information reaching the eyes and are thus a major driver of visual neural activity during natural visual behavior.

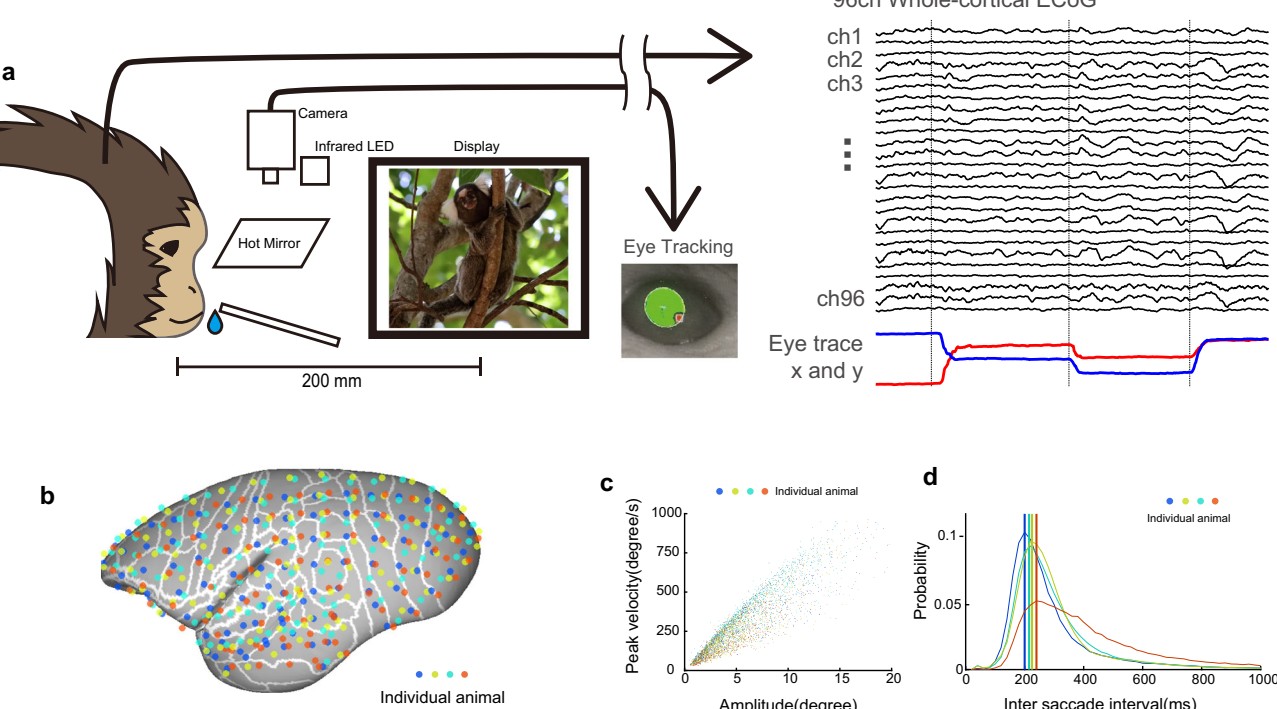

**Fig. 1 Large-scale ECoG recording of natural visual behavior. a** A schematic illustration of the experimental design. The marmoset viewed naturalistic movies while its eye movements and ECoG signals were recorded. A liquid reward was delivered at a random interval irrespective of behavior to keep the subject awake for a longer duration. **b** The electrode positions of the four subject animals. The ECoG array with 96 channels covered almost the entire lateral hemisphere of the marmoset brain. The electrode positions were determined by in vivo CT scans and the cortical areas inferred from registration of subject MR images to the marmoset atlas. **c** The main sequence of saccades during the free-viewing paradigm. **d** Inter-saccade interval. The marmoset made rapid eye movements every 220 ms during natural visual behavior. Color represents individual marmosets as in (**b**).

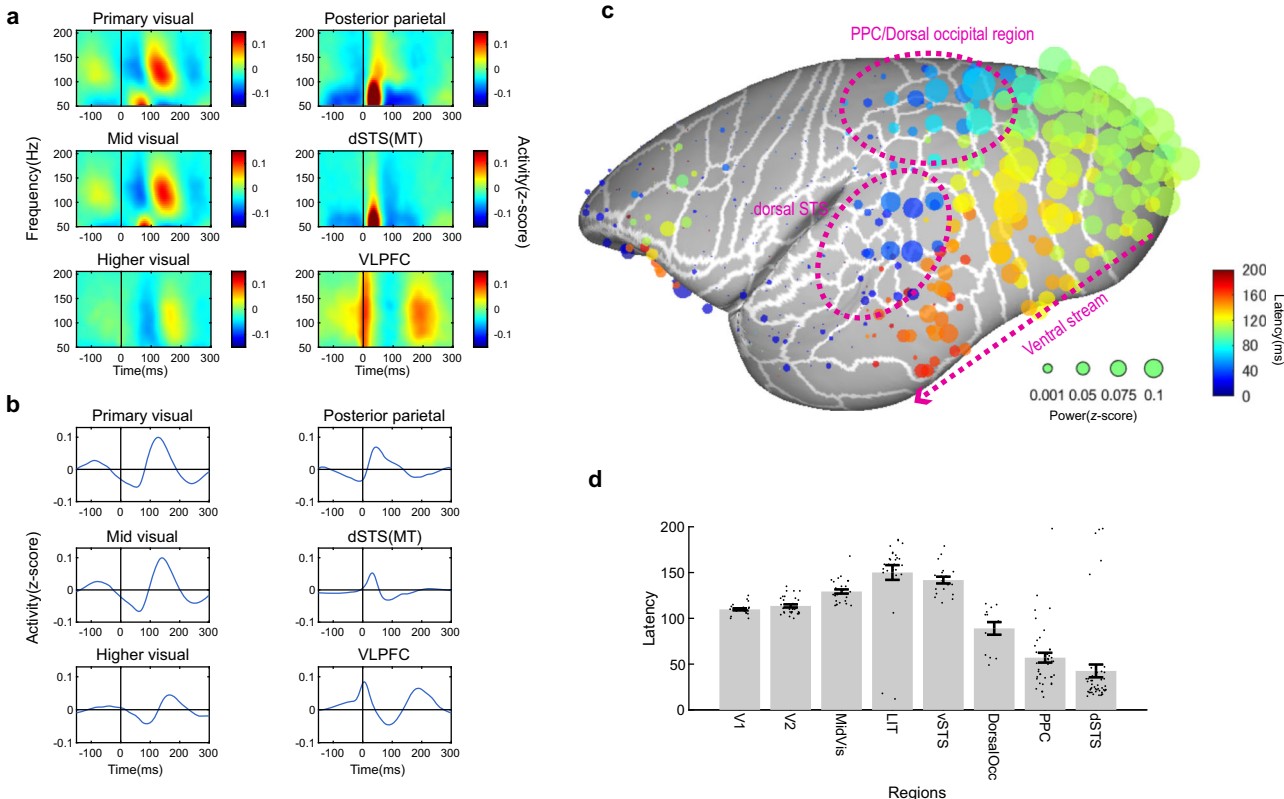

**Fig. 2 Perisaccadic activation pattern of active vision.** Post-saccadic neural activity of whole electrodes. **a** ECoG perisaccadic spectrogram of representative electrodes. The ECoG signals were aligned by saccade onset. Robust activity can be seen in visual areas after saccades. **b** High-gamma (100–160 Hz) ECoG signals in representative channels. **c** The magnitude and latency of the high-gamma signal peak from saccade onset. The data from four animals and 96 × 4 channels distributed over the entire lateral cortex (63 cortical areas) are plotted. The electrodes on the right hemisphere are computationally mapped onto the left hemisphere for visualization purposes. The smooth transition of visual information can be seen in the ventral stream. The areas in the dorsal stream (such as the dSTS including the MT, PPC, and dorsal occipital areas) were activated prior to the ventral stream. Statistical assessment of signal modulation by randomization is shown in Supplementary Fig. 1. **d** Mean latency of post-saccadic neural activity in each cortical region. Error bars showed standard error of the mean. Dots show individual electrodes. MidVis mid-visual areas (e.g., V3, TEO), LIT lateral inferior temporal areas (e.g., TE1, TE2), vSTS ventral superior temporal sulcus (e.g., PG/IPa, vFST), dSTS dorsal superior temporal sulcus (e.g., MST, MT, dFST), DorsalOcc. dorsal occipital areas (e.g., V6, V3a), PPC posterior parietal cortex (e.g., LIP, AIP, Opt). Supplementary Table 1 shows detailed area names included in each region of interest.

We first computed spectrograms from the ECoG signals (Fig. 2a) around saccade onset and then extracted the high-gamma band (Fig. 2b), which is highly correlated with local neural activity[24–26]. We found the smooth transition of post-saccadic activity along the ventral stream (Fig. 2c and Supplementary Fig. 1a), which was evident by the gradual shift of high-gamma activity after saccade from V1 (the most posterior part of the marmoset brain) to the anterior part of the temporal cortex (Fig. 2d). Conversely, the signal flow corresponding to the dorsal stream, i.e., the gradual shift of neural activity from early visual areas to temporoparietal regions, was less prominent (Fig. 2c). Instead, several cortical regions located along the dorsal stream showed a faster response than V1 (Fig. 2d). The fastest response was observed in the dorsal part of the middle temporal area (MT) and the dorsal part of the superior temporal sulcus (STS) (Fig. 2c, d), which we collectively termed the dorsal STS (dSTS). The second region that showed a faster response than V1 covered a large area, including the posterior parietal cortex (PPC) and dorsal occipital cortex (Fig. 2c, d).

Interestingly, high-gamma activity was suppressed prior to post-saccade excitation along the areas of the ventral stream (Fig. 3a, b and Supplementary Fig. 1b). This was less prominent in the regions of the dorsal stream, except for the dorsal occipital cortex (Fig. 3a, b). Surprisingly, the timing of suppression differed

drastically across the areas (Fig. 3c), i.e., it changed systematically according to the excitation timing of each area, so that the time difference between suppression and excitation remained constant, and the gradual shift of peak suppression could be seen along the ventral stream, as observed for the excitation signal.

These results suggest that in natural vision, in which saccades are critical for refreshing visual information on the retina, the dorsal stream plays a major role prior to V1, and then the ventral stream is activated for further visual analysis.

**Distinct patterns of motor- and visual-related signals in the frontal and the dorsal regions.** Perisaccade neural activity can be derived from various processes such as the reafferent visual signal from the eyes or motor-related signals, e.g., motor preparation[27] or efference copy/corollary discharge[28,29]. Here, we aimed to characterize further the potential differences across the dorsal areas that were activated prior to V1.

First, we tested contralateral dominance before saccade onset (i.e., −30–0 ms) as cortical saccadic motor control is known to be contralaterally dominated. We found that the ventrolateral prefrontal cortex (VLPFC) showed the strongest contralateral dominance among the entire cortex, and this was followed by the PPC and dorsal occipital regions (Fig. 4). This result indicated that the activation of these regions before or just after saccades is

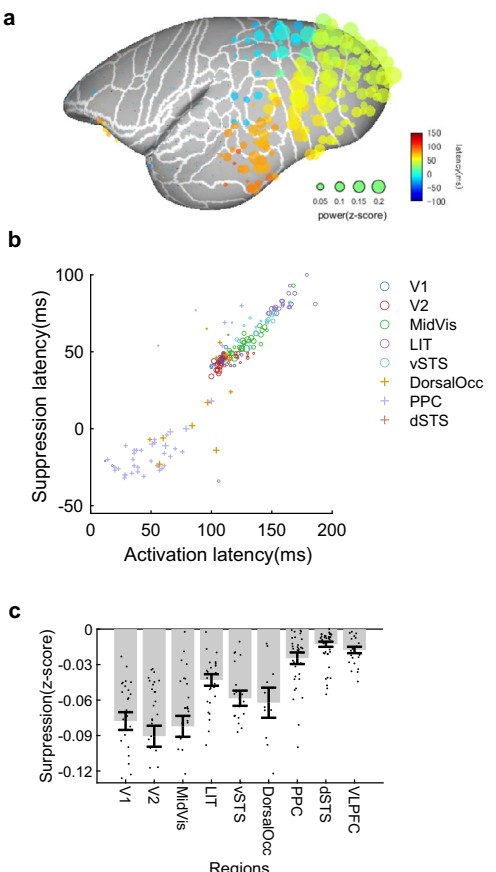

**Fig. 3 Presaccadic suppression pattern of active vision. a** Presaccadic suppression pattern across the whole lateral hemisphere from four animals. The size and color of the filled circles represent the magnitude and latency of the local minimum before saccade-evoked activation, respectively. The suppression magnitude of negative log *P*-values based on a randomization test is shown in Supplementary Fig. 1. **b** Correlation of activation and suppression timing across electrodes. Circles represent electrodes on the ventral stream, and the crosses are for the dorsal regions. A linear correlation can be seen between the latency of activation and suppression along the ventral stream. **c** Average suppression magnitude for each region. Error bars are the standard error of the mean. Dots show individual electrodes. The largest degree of suppression was observed across the entire ventral stream, and it was less prominent in the dorsal regions, except for the dorsal occipital cortices.

largely biased by saccade generation such as target selection and motor command. On the contrary, such contralateral dominance before saccades was virtually absent in the dSTS (Fig. 4). Moreover, the dSTS showed strong ipsilateral dominance from saccade onset, which contrasted with the VLPFC, PPC, and dorsal occipital regions. The VLPFC was activated before saccades and reached an activity peak at saccade onset (Fig. 2b), presumably because the signal is dominated by neural activity for the generation of eye movements. The second peak in this region was after ~200 ms, which is consistent with the typical visual response in primate prefrontal neurons[30]. Such a clear separation of the 1st and 2nd peaks was not evident in the dSTS and PPC (Fig. 2b). The activity peak of those regions occurred after eye movement was initiated (Fig. 2b). The dSTS and PPC are known to be involved in the control of eye movement[9,31,32], but the critical time window in which the dSTS and PPC function in natural visual behavior is quite distinct from that of the VLPFC.

Second, we attempted to disentangle the characteristics of the dSTS and PPC regions. Here, we examined whether the rapid

response just after saccades in these regions is derived from saccade onset or fixation onset. These two events are temporally close, but their impact on the retina is distinct, e.g., the former blurs the image, while the latter stabilizes it. In natural visual behavior, a variety of saccade amplitudes occur so that saccade duration ranges from 20 to 60 ms. This enabled us to test whether post-saccade activity was aligned by saccade onset or fixation onset (saccade offset). Interestingly, activation of the dSTS, including the MT, was strongly aligned with saccade onset, but not with saccade offset (Fig. 5a, b and Supplementary Fig. 2). Furthermore, suppression of the dSTS, which occurred after activation, was well explained by the timing of fixation onset, but not by saccade onset. This indicates that the dSTS was active when the eyes were moving and was suppressed when they stopped moving. The activity of the PPC contrasted with this pattern (Fig. 5a, b and Supplementary Fig. 2) and aligned well with fixation onset, while its suppression, which occurred prior to activation, was tightly aligned with saccade onset rather than fixation onset. This indicates that the PPC was suppressed while eyes were moving and was activated once they became stationary.

In summary, these results showed that the characteristics of cortical activation prior to ventral stream activation differed across regions among the frontal and dorsal areas. Activation of the VLPFC was dominated by contralateral saccades, indicating signal-related saccade generation. The dSTS was activated while the eyes were moving and suppressed once they were fixated, and the majority of PPC activity was suppressed while the eyes were moving and then activated at fixation onset.

**High-gamma signal trajectory accompanied by a traveling wave of theta oscillations.** Next, we sought to describe the spatiotemporal pattern of signal flow across the cerebral cortex. As high-gamma activity was transient and its timing differed across regions, it seems that the high activation spot should be limited in space and time. Figure 6a shows the high-gamma power of each time window around saccades. Indeed, the high-gamma signal was compact in space, and the activated area shifted gradually over time (Fig. 6a, b). The center of gravity of the active region traversed from the dorsal region, from anterior to posterior, and was eventually transmitted along the ventral stream (Fig. 6c and Supplementary Fig. 3), which contrasts with passive visual presentation, in which the signal generated in the occipital cortex was transmitted to the dorsal and ventral streams (Fig. 6c and Supplementary Fig. 4). The signal trajectory from the dorsal areas, which are usually placed in the dorsal stream, to the earlier visual areas (V1 and V2) is plausible as a substantial number of anatomical connections exist in this direction (Supplementary Fig. 5)[33]. Granger causality analysis also supported the presence of signal influence from the dorsal visual areas to the posterior part of the occipital visual areas (Supplementary Fig. 6).

Growing evidence suggests that synchronized neural activity is accompanied by low-frequency oscillations. The phase of these oscillations forms a spatial gradient on the cortical surface and travels as a wave along the cortex carrying information across distant regions, namely the traveling wave[34–36]. A recent study showed that the phase of the spontaneous traveling wave alters the visibility of subtle visual stimuli in marmosets[37]. Here, we tested whether the transition of the high-gamma signal peak observed in this study might be accompanied by such a traveling wave of lower frequency oscillations. We computed phase-amplitude coupling of the perisaccade period across the low-frequency phase and high-gamma signal amplitude. We found that the high-gamma signal was coupled with low-frequency oscillations of 6–12 Hz (Fig. 7a–c). Next, we analyzed the directionality of phases in this frequency range. We found that

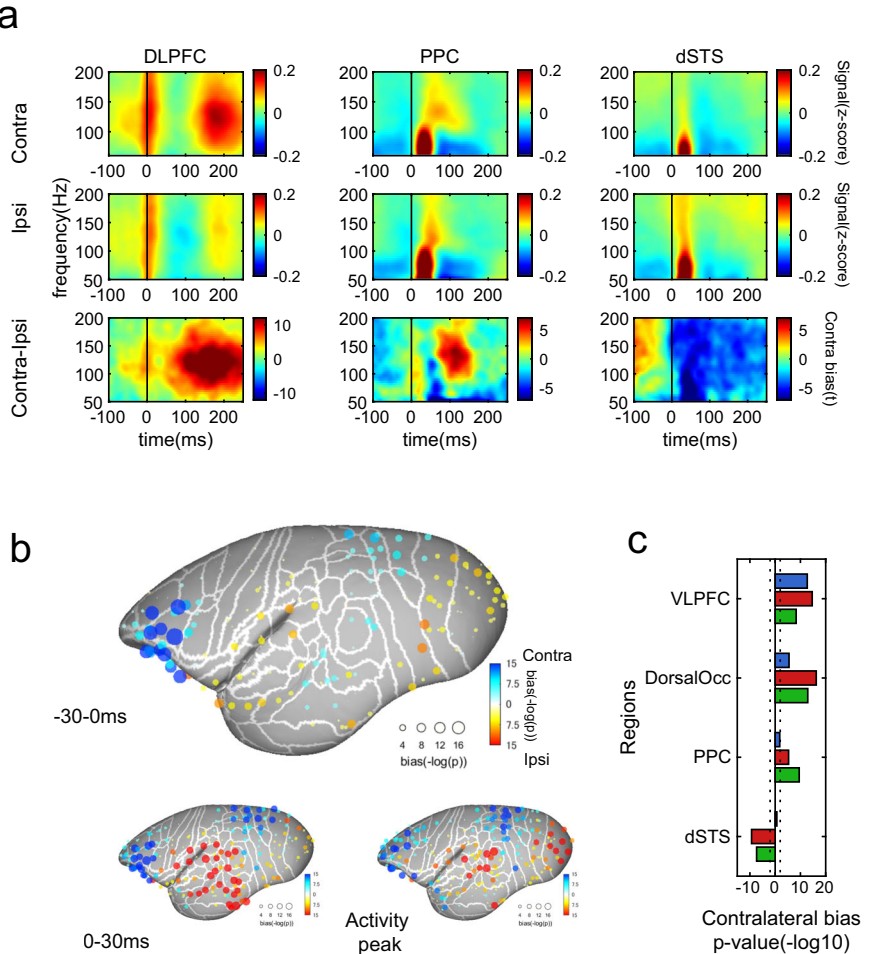

**Fig. 4 Contralateral dominance in the ventrolateral frontal cortex. a** Perisaccade spectrogram for an example electrode on the ventrolateral frontal cortex for contralateral and ipsilateral saccades. **b** The degree of contralateral dominance around saccade onset is shown by the color and size of the symbol on the 3D brain model. Blue color shows activity was larger for contralateral saccades. The largest degree of contralateral dominance appeared on the dorsolateral PFC (DLPFC) region. **c** Average contralateral dominance in each region of interest. Dotted lines represent statistical significance level with false discovery rate correction (alpha = 0.05).

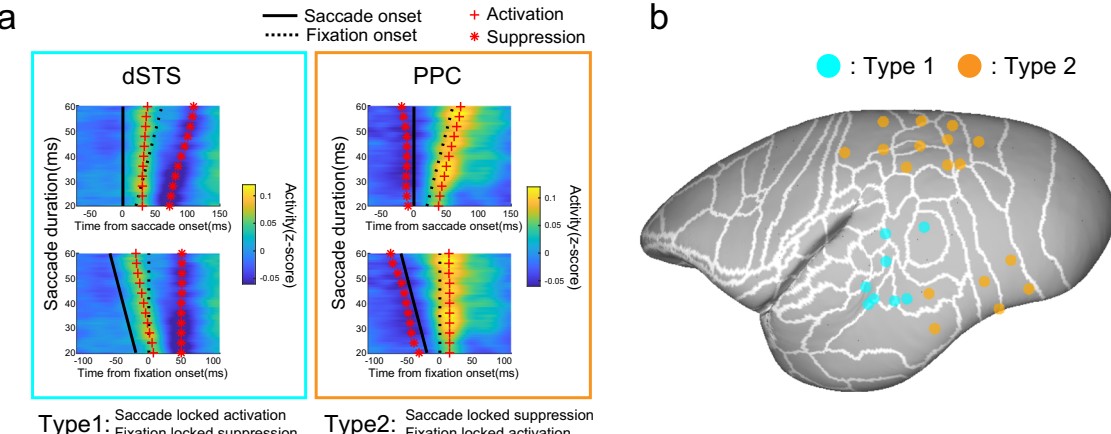

**Fig. 5 Distinctive profiles across the dSTS and PPC on saccade- or fixation-evoked activation and suppression. a** Examples of perisaccadic high-gamma signals aligned either to saccade or fixation onset. High-gamma signal modulation with different saccade durations was computed to test whether the signal was better explained by saccade (top row) or fixation onset (bottom row). The solid and dotted lines show saccade and fixation onset, respectively. The red crosses are the timing of the activation peak for each saccade duration, and the asterisks are for suppression. For the dSTS, activation timing was well explained by saccade onset, whereas suppression was according to fixation onset (type 1), and vice versa for the PPC (type 2). **b** Electrode positions either classified as type 1 (cyan) or type 2 (orange). Many type 2 electrodes were in the PPC region, while type 1 electrodes clustered around the dSTS.

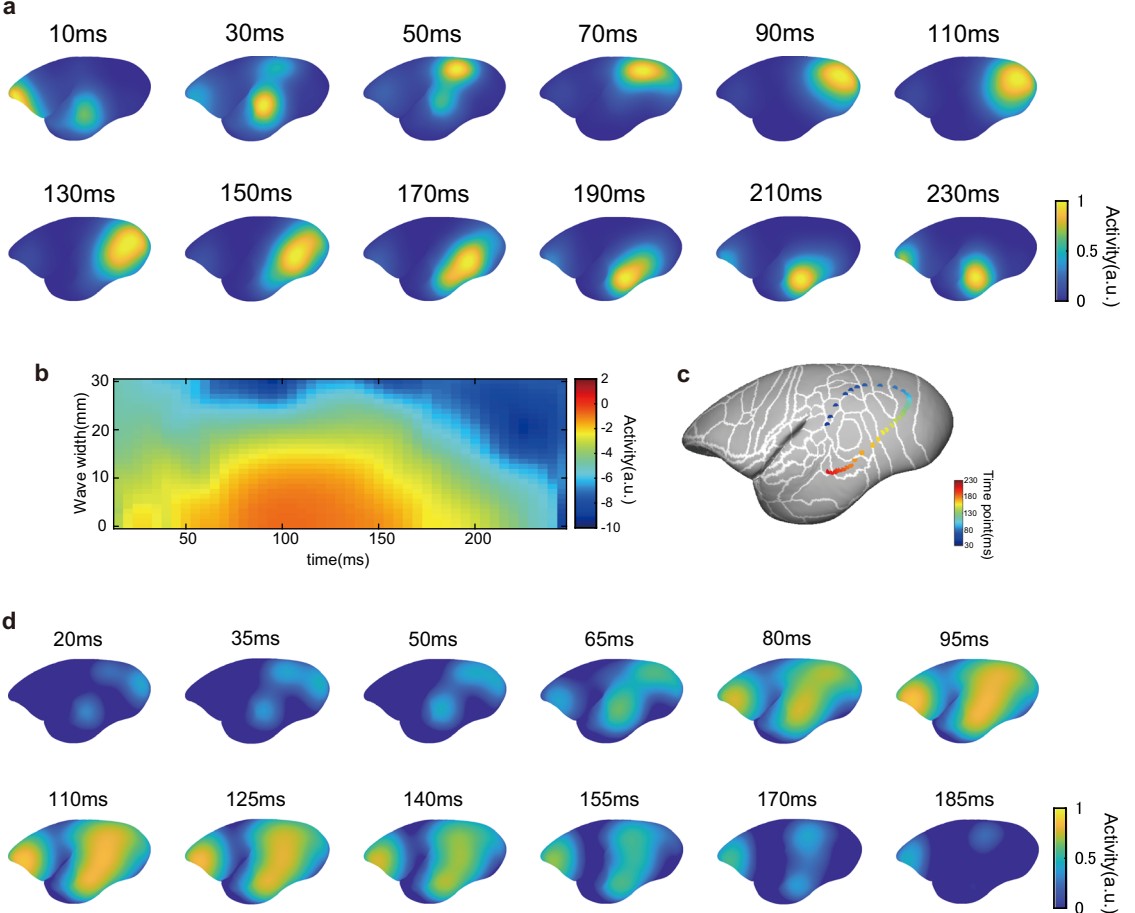

**Fig. 6 Spatial–temporal dynamics of high-gamma signal uniquely to active vision. a** Magnitude of the high-gamma signal in each time bin around saccades. Electrodes with a high-intensity signal were spatially clustered and moved across time. **b** High-gamma signal distribution across the cortical sheet as a function of distance from the center of high-gamma activity on the cortical surface in each time point. The high-gamma signal was spatially localized. **c** Center of gravity of the high-gamma signal. Colors represent the time from the saccade. The high-gamma signal peak moved from the dorsal region to the ventral stream. **d** Signal dynamics evoked by passive visual stimulus onset. In contrast to active vision, the initial response emerged at the most posterior part of the occipital regions and dSTS and this expanded toward the temporal and parietal cortices, which correspond to the ventral and dorsal stream, respectively.

the traveling wave propagated from the posterior terminal of the occipital cortex to the anterior pole of the temporal cortex along the ventral stream (Fig. 7d–f). Interestingly, in the frontal lobe and dorsal regions, the direction of the wave was reversed; namely, it propagated from the frontal pole to the posterior parietal regions. In addition, the wave around the dorsal occipital regions changed its direction of propagation to the ventral regions. This pattern of wave propagation was consistent with the trajectory of the high-gamma signal peak, suggesting high-gamma activity carried a traveling wave at a lower frequency range across the cortical surface from the dSTS, PPC, and dorsal occipital regions, and then to the ventral stream.

**The rhythm of brain dynamics linked to saccade behavior.** Such high-gamma activity propagation occurred 5 times per second, which corresponds to the rhythm of saccades in natural visual behavior. How is the cycle of the signal dynamics determined? When the neural activity of representative electrodes was aligned by saccade (saccade $i$), each area showed three distinct local peaks that consisted of the activity of saccade $i-1$, saccade $i$, and saccade $i+1$. This resulted in an interesting situation in which at least some cortical areas were always active between two adjacent saccades, and the signal was relayed from one area to another without any moment at which the activity of the entire brain

returned to background levels (Fig. 8a and Supplementary Fig. 7). It seems that the saccades are timed to occur before the activity of the whole brain returns to pre-activation levels and occurs at an interval that is sufficiently long to ensure that the signal reaches a variety of cortical areas.

To test this hypothesis directly, we computed the total activity of the whole brain and compared it with the inter-saccade interval. We found that a saccade occurred most frequently at ~220 ms after the previous saccade, and this timing coincided with the total activity of the whole cerebral cortex returning to background levels (Fig. 8b and Supplementary Fig. 8). In this manner, cortical computation is maintained at a certain level of activity without being completely inactive, and thus the neural network and behavior are optimized for the efficient computation of a visual scene.

**Parallel saccade generation at various visual processing stages.** Our observation of the correspondence between the timing of saccade generation and cortical activity returning to background levels raises two possible explanations for how saccades are generated during natural visual behavior. First, a saccade occurs when the analysis of a visual scene with a new eye position has been completed along the entirety of the visual stream. Alternatively, any part of the sensory cortical hierarchy could influence

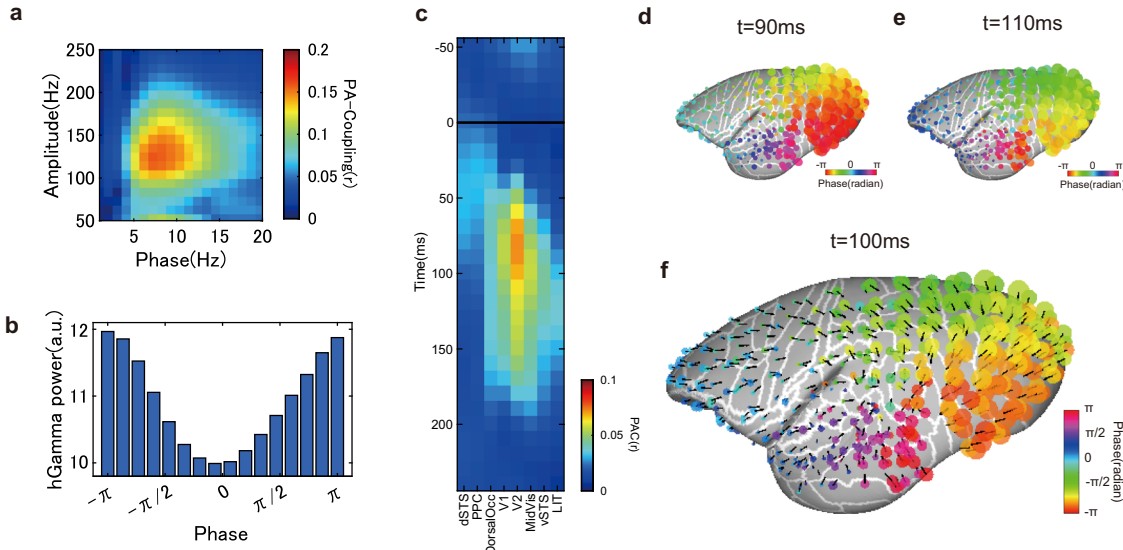

**Fig. 7 High-gamma signal coupled with a traveling wave of theta oscillations. a** Phase-amplitude coupling (PAC) of the perisaccadic period from an example electrode. High-gamma signal amplitude correlated best with the phase of theta and alpha oscillations (6–12 Hz). **b** High-gamma signal relative to the phase of theta oscillations (8 Hz) in the same electrodes as in (**a**). The High-gamma signal is highest at ±π rad and lowest at 0 rad. **c** Perisaccadic theta phase–high-gamma amplitude coupling across areas. Each column represents the average PAC from multiple electrodes belonging to the same cortical region. The solid black line is saccade onset. **d**, **e** Phase of theta oscillations at 90 and 110 ms from saccade onset. The phase of theta oscillations showed a gradient across adjacent electrodes. The phase of two adjusted time points shows the direction in which the phase moved along the cortical surface. **f** Directionality of traveling wave propagation. The black thick lines represent the direction of the traveling wave (the direction in which the same phase appeared at the next time point). The pattern of wave propagation was comparable to that of a high-gamma signal trajectory.

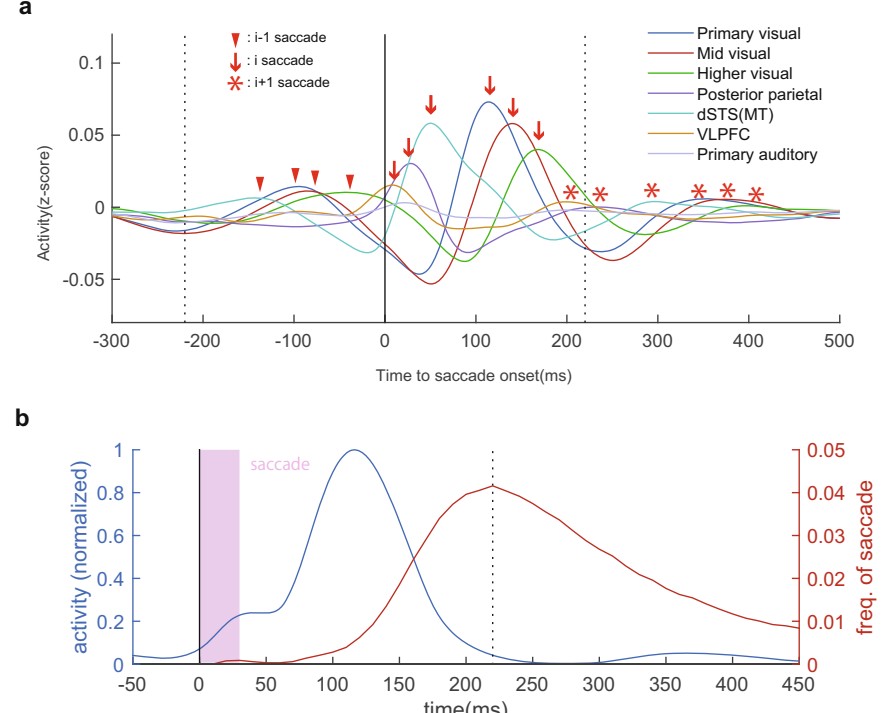

**Fig. 8 Recurrent dynamics of neural activity and saccade behavior. a** Relay of information across cortical areas around a saccade. Red arrows, representing the local peak activity of each electrode, show three peaks that were derived from the $i-1$, $i$, and $i+1$ saccade, respectively. Local maxima of high-gamma activity were successively relayed across cortical areas during natural vision, and the cortex did not return to baseline levels of activity as the next saccade occurred every 220 ms. **b** Inter-saccade interval and temporal dynamics of whole-brain activity. Whole-brain activity almost returned to background levels at 200–250 ms after the saccade, and this timing corresponds to when the next saccade emerged most frequently.

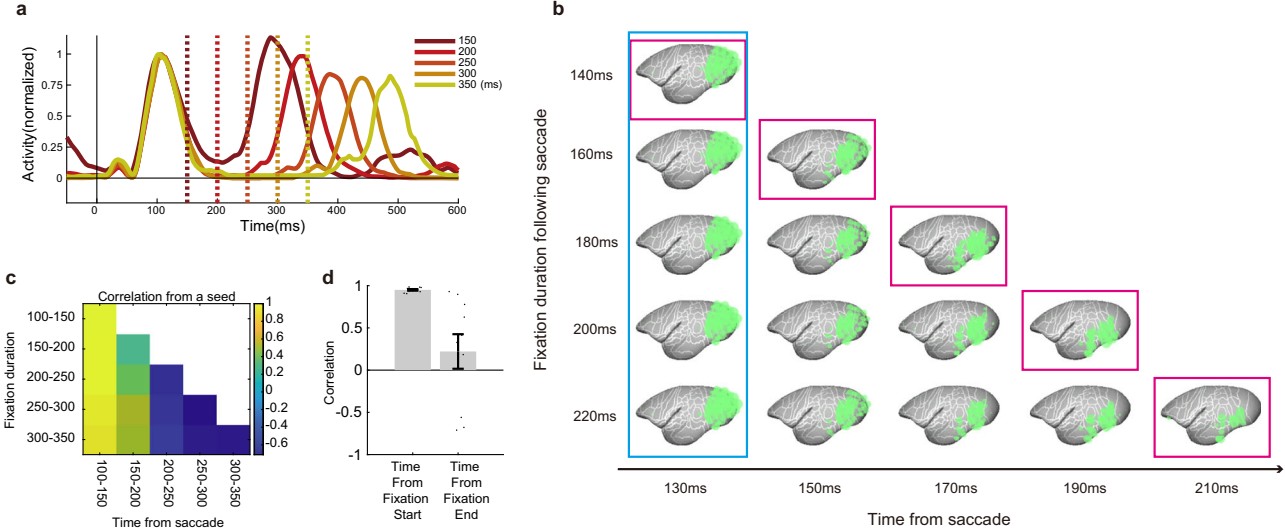

**Fig. 9 Parallel computation for saccade generation: distinctive activity profiles at fixation termination. a** Time course of whole-brain activity at different fixation durations. Regardless of fixation duration, which is the time available for determining the next saccade, brain dynamics were consistent (0–200 ms after saccade onset), i.e., the timing when whole-brain activity returned to background levels remained at ~200 ms. **b** Activity patterns for saccades with different fixation durations. Even when fixation duration was very short, the activity wave did not propagate rapidly across the cortex; instead, the activity dynamics were very similar within the same time range from saccade, regardless of the following fixation duration (blue box). On the contrary, the activity patterns when fixation terminated were drastically different across fixation duration (magenta box). **c** Example of the correlation matrix of the cortical activity fingerprint. The seed was the top-left image of (**b**). Again, activity was similar at the same time range after the saccade, rather than the time before fixation termination. **d** Mean correlation for activity in the same time range just after the saccade (blue box in b) and in the same time range just before fixation termination (magenta box). Dots show individual pairwise correlation The brain state of fixation termination that is drastically different according to time can be used to determine the next saccade target ($t(18) = 3.6$, $P < 0.001$), suggesting parallel computation of saccade target.

saccade generation in parallel, for example, via direct projections to the midbrain, the superior colliculus[38]. In this view, the correspondence between saccade timing and the timing of cortical silencing is simply a balance point between parallel saccade generation from different cortical computational stages. That is, a saccade may sometimes be generated by an earlier part of the cortical hierarchy or at the final stage. However, on average, the rhythm of active vision is designed to be neither too long so that the entire cortical computational process returns to background levels nor too short so that many cortical regions are unable to contribute to determine the next saccade.

To test these hypotheses, we examined how activity patterns changed according to fixation duration before the next saccade, which is the time that can be used for visual computation of the current eye position. Figure 6a shows the saccade-evoked cortical-wide signal dynamics followed by different fixation durations. The signal peak after the termination of fixation (the 2nd local peak from the saccade) was systematically delayed according to fixation duration, which confirmed our analysis worked well as the 2nd peak is derived from the subsequent saccade that terminates fixation, and this timing was delayed according to fixation duration. Conversely, we found that cortical signal dynamics were very similar after a saccade until fixation termination, regardless of fixation duration (Fig. 9a and Supplementary Fig. 9). In turn, when the fixation period was short, the activity level of early cortical areas remained high, even just before saccade onset, and the higher visual areas did not reach their activity peak (Fig. 9b and Supplementary Fig. 10). Furthermore, the activity pattern of this fixation duration was very similar to the activity pattern for longer fixation durations (Fig. 9c, d). This suggests that the speed and pattern of a signal traversing the cortical sheet did not change according to the subsequent fixation duration. Conversely, the pattern of cortical activity just before the saccade was drastically different according to fixation duration before the saccade (Fig. 9b, magenta boxes;

Fig. 9d). When fixation duration was short, the anterior part of the inferior temporal regions did not receive full visual information, and only the early and mid-visual areas reached a high activation level before the next saccade target was selected. Conversely, when fixation duration was long, the anterior part of the temporal cortex tended to increase its activity toward saccade generation (Fig. 9b). These results support the view that the visual cortices are able to contribute to the generation of saccades at any stage of cortical visual computation in parallel.

In Fig. 9a, there appears to be a relationship between fixation duration and the power of subsequent peaks. We are not sure for the precise reason why the power of subsequent peaks differed across fixation duration. The signal pattern after fixation initiation (0–200 ms; 0–130 ms for the 150 ms fixation condition) was quite similar regardless of how long the subsequent fixation was maintained, however, the activity profiles just before the next saccade differed drastically according to the prior fixation duration. This difference might explain the difference in the power of the subsequent peak.

It should be noted that the absence of cortical activity before saccade onset shown in Fig. 7b cannot be attributed solely to saccade suppression[39]. If this was the case, then the timing when cortical activity returned to background levels should systematically shift according to the timing of the next saccade; however, this was not supported by our results (Fig. 9a and Supplementary Fig. 9).

## Discussion

In this study, we identified several distinctive features of brain dynamics during active visual behavior captured by an ECoG array covering almost the entire lateral surface of the marmoset cortex. During the course of active visual behavior, we found that the dorsal stream acts even earlier than V1, and activation of the ventral stream follows. The high-gamma signal traverses the

cortical sheet from the dorsal to ventral areas accompanied by a traveling wave of theta oscillations. This signal was generated five times per second, just before the activity of the entire brain returned to background levels. We further demonstrated that saccades could be generated at any step of visual computation, indicating the parallel computation of motor commands. In this manner, neural network architecture and visual behavior are coordinated for the efficient exploration of the environment.

Primate eyes sample a visual scene recursively with different eye positions, and the timing and target of the next saccade are determined by the results of ongoing visual processing. Our study showed, in the free-viewing condition, the VLPFC plays a prominent role in triggering saccades, and visual information, while eyes are moving, may first arise in the dSTS. Area MT (in the dSTS) has a neural pathway that bypasses the primary visual area to receive visual input from the eyes[11,14], and can thus receive visual information even faster than V1. Rosa and Tweedale[40] claimed that the MT can be thought of as "an additional primary visual area" as it has strong direct input from the lateral geniculate nucleus[41]. In this sense, V1 is not a unique cortical area that receives the earliest visual input from subcortical structures. This information may merge with the signal in the PPC and dorsal occipital cortices, presumably involving computation for the spatial conversion of visual information across different eye positions[29]. Then, with a slight delay to this process, information flows along the ventral stream for further analysis of object semantics. At approximately the time when the anterior part of the temporal cortex receives visual information, which coincides with the second peak of frontal cortical activity, the next saccade will likely occur before the whole-brain computation process becomes silent. This represents a cyclic flow of information that fits the recursive and dynamic nature of visual behavior.

Our findings raise several questions that remain to be investigated. First, what is the neural basis for the distinctive timing of saccade suppression along the ventral stream? The detailed mechanism for saccadic suppression is still controversial, but one of the most plausible hypotheses assumes that the visual areas might receive extra-retinal information that indicates the timing and profile of saccades from the areas controlling saccadic eye movements such as the superior colliculus[42]. However, solely with these models, the differential timing of saccadic suppression along the ventral stream might not be explained. Second, what computational process underlies the distinctive activation pattern of perisaccadic periods across the dSTS, PPC, and VLPFC? At mesoscopic levels of ECoG recording, we found that the dSTS was active while the eyes were moving and suppressed at fixation onset, and vice versa for the PPC. We speculate that the signal in the dSTS is dominated by the reafferent signal from the eyes, as ipsilateral dominant activity was seen just after saccade onset in these electrodes (Fig. 4). Eye position was not controlled in our experiment, so gaze position on saccade onset was random across the stimulus movies; however, ipsilateral saccades, on average, tended to bring the stimulus inside the receptive field of neurons under the dSTS electrode. This view is consistent with a previous study of single-unit recordings of macaques showing that MT/ medial superior temporal neurons retain stimulus-evoked activation during saccades, and the receptive field moves along with the trajectory of saccades on the stimulus display coordinates[43]. The activation and suppression patterns of the PPC are apparently counterintuitive. The lateral intraparietal region in the primate PPC is known as the parietal eye field that controls saccadic eye movement. This apparent discrepancy might be explained by that fact that the motor receptive field differs across neurons covering the entire lateral hemifield as a population, and they are suppressed if saccades are away from the receptive field. Thus, as ECoG signals represent a summation of multiple

neurons, the high-gamma signal was suppressed before saccades and activated at fixation onset. The signal increased rapidly with fixation onset, which was stronger for contralateral saccades and this bias was extended further during fixation. This signal might include a mixture of the eye-position signal, efference copy, and visual information; however, it is difficult to come to a conclusion on this issue with the findings of the present study. A combination of saccade experiments under complete darkness or the precise control of eye position is necessary to disentangle further the precise content of the signal and its interactions across those areas in natural vision.

In any case, it is worth noting the signal dynamics of natural active visual behavior. The visual system is one of the most investigated cortical circuitries, and an enormous number of studies have characterized the functional properties of each region. Many different behavioral tasks have given different functional labels to single cortical regions, which makes it difficult to extract the principal computation of each region. Therefore, a precise description of the timing of activation of cortical areas and which regions exchange signals under natural behavior might provide unique insights for the construction of a visual computation model.

Not only primate visual behavior but also many sensory systems, such as mice whisking and sniffing samples, occur rhythmically[44], while the rationale for such oscillatory processes is elusive[45,46]. The present study provides a simple view of how this timing is regulated. As the visual system is organized hierarchically, it takes a certain amount of time until the final visual center has completed the analysis of new visual information, and then the visual system can decide which direction the eyes should be moved in. This network property may simply determine the interval of visual exploration, and thus the rhythm of visual exploration is intrinsic to the hierarchical neural architecture.

Note that we do not assume that saccade generation is a completely serial process, rather we believe it is a parallel computational process. Indeed, many saccades are generated far earlier than when the higher visual areas receive the information for a new eye position. In this case, the saccade target is determined without the full contribution of the higher visual areas. In fact, a saccade following short fixation is influenced more by visual salience than a saccade after long fixation, probably due to faster computation within the superior colliculus than for semantic analysis in the ventral stream[47]. Guillery and Sherman[38] claimed that any of the steps of cortical sensory computation can send a certain type of instructor signal to the motor apparatus; on the basis of neuroanatomical evidence, they further conceptualized such circuitry in which motor instructions are sent from multiple cortical areas in parallel at different levels of the cortical hierarchy. Our data, showing a distinctive activity profile at fixation termination, support this view. Along with this theory, it might be fruitful to disentangle the information dynamics according to saccade parameters such as prior fixation duration or the visual features of the target that triggered the saccade. Schall et al.[48] proposed a beautiful schema showing an anatomical connectivity gradient in the anterior bank of the arcuate sulcus that coexists with a functional gradient in saccade amplitude at the frontal eye field. The visual system is designed to work under natural behavior, so there must be a reasonable link across different types of saccade (amplitude, interval, and target visual features), its underlying neural architecture, and the visual environment. Our current analysis focused on neural dynamics of the trial averaged signal. The analysis with a single-trial level may reveal such a link across behavior, neural architecture, and environment in future work.

Our results demonstrate only the most prominent route of signal flow by ECoG recording, which represents the summation

of the activity of the heterogeneous neural population on the cortical surface under each electrode. To differentiate information dynamics with a finer resolution by ECoG, further technical challenges need to be addressed. For example, the magnocellular and parvocellular layers in the lateral geniculate nucleus represent functionally distinct information, and these convey information to succeeding areas while keeping a certain degree of segregation across channels[49–51]. In this study, we did not track these two streams independently. Similarly, signal transmission from early visual areas to the dorsal stream (as expected from the dual-stream model) could not be identified in the active exploration condition (while it was supported by Granger causality analysis), and was only observed in the passive-viewing condition. The activity of all heterogenous neurons in the dorsal regions was mixed in a single channel in our analysis, and this may have prevented us from observing signal transmission from V1 to the dorsal stream because of the dominance of the faster visuomotor response just after saccades (Fig. 4a). Similarly, our study primarily focused on the high-gamma signal as this frequency range is known to be highly correlated with local neural activity; however, signals in different frequency ranges are likely to convey different types of information that are complementary to the high-gamma signal[52]. Future studies require technical advances such as a combination of ECoG and circuit-specific genetic manipulation[53–55] to capture the information stream with a finer resolution and to provide further details for the model of visual information dynamics in active vision.

## Methods

**Ethics information**. All procedures in the animal experiments were approved by the Wako Animal Experiment Committee (Animal Care and Use Committee), RIKEN, and all experiments followed the institutional ethical guidelines for the care and use of experimental animals.

**ECoG implantation**. The ECoG array, consisting of 96 electrodes, was chronically implanted in a half hemisphere of four marmoset monkeys (Callithrix jacchus) (two each for the left and right hemispheres) following the protocol developed by Komatsu et al.[19]. The contact area of each gold-plated copper electrode was 0.8 mm in diameter (Cir-Tech Co., Ltd.). We performed a craniotomy of ~2 × 2 cm and gradually inserted the ECoG sheet between the skull and dura. The reference electrode was placed on the dorsal part of the somatosensory cortex on the contralateral hemisphere, and the ground electrode was attached to the skull of the contralateral hemisphere. About 6–8 plastic screws (1.4 × 2.5 mm) were implanted in the skull, and a connector of the electrodes was fixed with the skull and the screws using acrylic cement. A plastic headpost was also attached to the skull beside the connector. The marmoset was immobilized by ketamine(15 mg/kg) and maintained under anesthesia using isoflurane (1–3%) during all surgical procedures. The condition of the animal was monitored by body temperature and arterial blood oxygen saturation. Each surgery was conducted under aseptic conditions.

**Electrode localization**. We localized the electrodes using computed tomography (CT) and magnetic resonance (MR) imaging. We acquired T2-weighted MR images in advance of ECoG implantation. The marmoset was anesthetized and maintained by 1–3% isoflurane during imaging while their body temperature, heart rate, respiration, and SP0₂ were monitored. T2-weighted MR images were taken using a RARE sequence with the following parameters: TE = 11 ms, TR = 4,000 ms, FA = 90°, RARE factor = 4, matrix size = 178 × 178, slice number = 48, resolution = 0.27 × 0.27 × 0.54 μm. CT images were taken at 1 week after ECoG implantation when the animal had recovered fully. The animal was anesthetized by ketamine (30 mg/kg i.m.) while its respiration was monitored. CT images were taken at an isotropic resolution of 60 μm. To infer electrode positions on the brain surface, the CT images were aligned manually to the T2-weighed images using AFNI software[56] (http://afni.nimh.nih.gov). The location of the electrodes was segmented manually on the CT images using 3D Slicer[57] (http://www.slicer.org). To help annotate the regions where the electrodes were attached to the brain surface, the cortical annotation of the marmoset MR imaging atlas[58] was transformed onto the subject space by using a free-form deformation named Symmetric Normalization, implemented in the ANTs toolbox[59]. We determined the putative cortical areas of the electrodes based on visual inspection of the MR images compared to the standard atlas[60] and digital annotation based on spatial deformation on the subject brain.

**Behavioral paradigm**. Four marmoset monkeys observed 18 variations of 10-min movies that contained a variety of naturalistic scenes (such as social interactions of monkeys). To maintain the arousal level of the marmoset, a small amount of liquid reward was delivered while movie viewing. In the passive-viewing task, static images, which were from the same movies of the free-viewing task, were presented sequentially to the marmoset with an average stimulus interval of 1200 ms. Reward timing was distributed randomly with a mean interval of 1500 ms and was independent of gaze behavior. The marmoset viewed 2–4 movies per day. The viewing distance was ~20 cm and the stimulus had a visual angle of 40 × 22.5°. Stimulus and reward timing was controlled by a custom-written MATLAB program using Psychtoolbox[61,62] (http://psychtoolbox.org) and synchronized with the ECoG signal by an analog trigger signal via the National Instrument DAQ system. Eye movement was obtained by pupil-corneal reflection methods using an infrared camera (GS3-U3-32S4; Point Gray) at 500 Hz with iRecHS software[63]. We performed gaze calibration using a protocol based on the one described by Mitchell et al.[64]. In short, small images of interest to the marmoset were presented on the display for 7 s, and this was repeated with a different stimulus set for a total of five times. Four parameters adjusting gain and shift in the x and y directions were estimated to maximize their gaze to be on one of the images as they spent more time viewing those stimuli than blank space on the display.

**ECoG recording**. The ECoG signal was recorded at 1 kHz with a bandpass filter ranging from 0.3 to 250 Hz using a data acquisition system (Ripple; Grapevine). The signal was digitalized and multiplexed at the head stage using the common reference, which was placed on the foot region of the somatosensory cortex on the contralateral hemisphere. We analyzed 79 movie viewings with approximately 137,000 saccades, 70 with 135,000, 70 with 128,000, and 63 with 81,000 for each animal, respectively, for the free-viewing task. For the passive-viewing task (Supplementary Fig. S1), we analyzed approximately 40 sessions per animal including 7680 stimulus presentations. We performed this task in two of the four animals with implanted electrodes.

**Saccade analysis**. Saccades were extracted by an acceleration filter and logistic fitting to eye-movement data following the protocol proposed by Mitchell et al.[64]. Raw eye traces were smoothed by a median filter (100 ms window size) and a second-order Butterworth noncausal filter (−3dB at 50 Hz)to reduce high-frequency noise. Then, the candidates for saccade onset and offset were extracted by the velocity(over 10 degrees/s) and acceleration (over 1000 degrees²/seconds) profile of the smoothed eye trace. For each candidate saccade, we fitted the logistic function to the eye trace of the perisaccadic period and compared that with a spline model having the same number of parameters. The logistic model consisted of three parameters fitted to the mean, linear and quadratic trends over 150 ms time series of eye-position data, centered by a pair of candidate saccade onset and offset. The other two parameters fit the width and the amplitude of the logistic function. The spline model was fourth-order spline with an additional parameter for the mean. We consider a saccade only when logistic fitting explained the variance of the perisaccadic eye trace 50% better than that of the spline model.

**ECoG spectrogram analysis**. To create spectrograms from the ECoG data, we applied a bandpass filter to the raw ECoG signals and then obtained the envelope by applying the Hilbert transformation as the signal intensity of each frequency band. Bandpass width was 4 Hz and center frequency was moved from 4 to 200 Hz. The resulting time courses of the envelope were z-scored in each frequency range and in each channel, and then this was highpass-filtered to remove signal drift (cutoff = 0.1 Hz). High-gamma signal intensity was obtained as an average across 100–160 Hz, which is known to be highly correlated with local neural activity[24–26]. The magnitude and latency of perisaccadic activation peak were obtained as the local maximum from 0 to 250 ms from saccade onset. This was obtained from the averaged signal time course aligned by saccade onset. Saccadic suppression was a local minimum before saccade-evoked activation for each channel.

**Contralateral dominance and activity with different fixation durations**. To estimate the activation pattern of ipsi/contralateral saccades, we performed the same analysis as above, except that only a subset of saccade onset was used to obtain the averaged signal. Contralateral dominance was determined by subtracting the perisaccadic activity of ipsilateral saccades from that of contralateral saccades. Subtraction was performed using the z-scored spectrogram data. The estimation of p-values is described in the Statistical Analysis sub-section. Similarly, in Fig. 9, we subsampled the saccades based on the duration of the subsequent fixation.

**Activation/suppression driven by saccade or fixation onset**. To determine whether the perisaccadic signal was better explained by saccade or fixation onset, we obtained two different average signal time courses aligned by saccade or fixation onset. We computed them with different saccade durations ranging from 20 to 60 ms (bin width = 3 ms, sliding step = 1 ms). To classify the channels to either type 1 (eye-movement type: activation by saccade onset and suppression by fixation onset) or type 2 (fixation type: suppression by saccade onset and activation by fixation onset) in Fig. 5, we performed the following analysis. First, we obtained the activation peak in different conditions (saccade/fixation onset × saccade duration),

and obtained the activation peak and suppression peak timing as local maximum/minimum. In each channel, we considered either only pre- or post-suppression whose signal modulation was larger than the other. Then, we performed regression of those latencies separately to the aligned event timing (saccade activation peak aligned to saccade onset, and fixation aligned to fixation onset, which are the same as for suppression). We obtained a saccade-fixation (s-f) bias by subtracting the regression coefficient for fixation onset from that of saccade onset for each activation and suppression regression coefficient. The s-f bias represents the extent to which degree perisaccadic modulation is better explained by saccade onset (a negative value indicates modulation is better explained by fixation onset). We considered an electrode as type 1 (eye-movement type) if the s-f bias for activation was greater than 0.4 and less than −0.4 for suppression and also that suppression was larger after activation. An electrode was defined as type 2 (fixation type) if the s-f bias for suppression was greater than 0.4 and less than −0.4 for activation and also that suppression occurred before activation. The original s-f values for activation and suppression are shown in Supplementary Fig. 2. We considered only electrodes whose regression coefficient was obtained reliably, i.e., the residual peak latency time as compared to estimated timing by regression was less than 10 ms on average across different saccade durations.

**The trajectory of the high-gamma signal**. To visualize the trajectory of high-gamma activity, we used a 3D vertex of the standard marmoset brain model. We assigned the high-gamma activity of each electrode on the nearest vertex of the 3D brain surface derived from nonlinear registration across the atlas and subject MR images as described in the previous section. Signal amplitude was derived from average activity aligned by saccade onset as described earlier. Then, we smoothed the activity power along the cortical surface using the "SurfSmooth" function implemented in AFNI software[56] (http://afni.nimh.nih.gov) with 10 mm FWHM. To assess the center of gravity of high-gamma activity, we computed the Fréchet mean of high-gamma activity from all ECoG electrodes as the vertex $p$ that was obtained by the following formula:

$$\underset{p}{\mathrm{argmin}} \sum_i w_i \cdot d^2(p, x_i) \tag{1}$$

where $x_i$ is a location of an electrode, $w$ is the high-gamma signal of the electrode, $p$ is any vertex of the 3D brain model, and $d(p,x_i)$ is the distance (geodesics) between $p$ and the electrode $x_i$. Geodesics between electrodes $x$ to an arbitrary vertex $p$ on the 3D brain model were computed using the MATLAB (MathWorks) utility "Exact geodesic for triangular meshes."

**Phase-amplitude coupling (PAC) and the traveling wave of theta oscillations**. PAC was computed as a circular-linear correlation[65]. When evaluating the time-evolving PAC of the perisaccadic period, we adjusted the size of the sliding window based on the phase frequency so that it contained one cycle, e.g., for 8 Hz theta oscillations, the sliding window size was 125 ms. PAC was computed trial-by-trial, i.e., without averaging phase or amplitude across trials; all time points within the window from all trials were concatenated to compute the circular-linear correlation. In the traveling wave analysis for Fig. 6, we analyzed the phase of theta oscillations (8 Hz). To determine the propagation direction of the traveling wave, at time $t$ and electrode $i$, we sought the electrode $j$ that minimized:

$$\left| P_{j,t+1} - P_{i,t} \right| \tag{2}$$

where $P$ represents the phase of electrode $i$ at time $t$. Then, the direction of $p_i$ to $p_j$ was computed on the surface model. In this case, phase was obtained by signal time course averaged across multiple saccades for a reliable estimation of theta oscillation phase at each timing of each electrode.

**Temporal dynamics of whole-brain activation and the effect of subsequent fixation duration**. Figure 8b shows a simple summation of the positive part of high-gamma activity for all electrodes from all four animals. The high-gamma signal was averaged time course aligned by saccade onset obtained as described above. Here, we considered only the positive component over baseline activity from all electrodes because the primary focus of the present study was to see to what extent the cortical activity evoked by a saccade remained before the next saccade was likely to occur. In addition, the inclusion of the negative component may underestimate the ongoing activation in other electrodes (see Supplementary Fig. 8 in the case when the negative component was included).

**Granger causality**. We assessed the degree of signal interaction and its directionality by Granger causality analysis, which is a statistical measurement indicating the extent that one time series with a slight time delay can predict another time series[66]. First, we computed the Bayesian information criterion (BIC) to the autocorrelation of a single channel, which is the regression equation to predict the time series of high-gamma activity of channel $i$ based on the past activity of the same channel $i$. Then, we performed the same process for regression-predicting channel $i$ from the past activity of channel $j$. We computed ΔBIC as the subtraction of the BIC for the autoregression model ($i$ and past $i$) from that of a pair of electrodes ($i$ and past $j$). Lower ΔBIC values indicate a higher degree of Granger

causality from channel $j$ to channel $i$. We repeated this process for all combinations of electrodes for each animal.

**Statistics and reproducibility**. For Figs. 2c and 3a, statistical significance was obtained by a randomization test in which saccade onset and ECoG signal were derived from different sessions of the same animal. Then, we obtained activation/suppression magnitude from the pseudo-randomly generated perisaccadic signal. We repeated this procedure 500 times, and the $p$-value was estimated based on the distribution of the resulting magnitude derived from randomization. We computed the P-value by fitting the Gaussian model to the resulting distribution, rather than directly estimating the P-value. This is because it was computationally too heavy to repeat randomization for the thousands of iterations that are required for the direct estimation of the P-value from the distribution. The false discovery rate for multiple comparisons was controlled by Storey's method[67]. The resulting P-values are shown in Supplementary Fig. 1.

For Fig. 4, we performed a randomization test to obtain statistical significance, where labels of contralateral or ipsilateral saccade were randomized within the same session, so that the null hypothesis is the signal pattern should be the same regardless of saccade direction. The following procedure was the same as described above.

In Fig. 9d, the similarity of the spatial–temporal pattern of cortical-wide activity was compared across "time from fixation onset" and "time from fixation end." The correlation value was computed from $96 \times 4$ channels from four animals in each time window with different fixation durations, and those values were compared across conditions ("time from fixation onset" and "time from fixation end") by an unpaired two-tailed $t$ test.

**Reporting summary**. Further information on research design is available in the Nature Research Reporting Summary linked to this article.

## Data availability
Source data underlying Figs. 2d, 3c, 4c, and 9d is presented in Supplementary Data 1. Other data acquired for this study is available upon reasonable request.

## Code availability
The custom code used for data analysis in this study is available upon reasonable request.

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

## Acknowledgements

We thank Yuri Shinomoto for animal care, training, and awake recordings; Dr. Naomi Hasegawa for veterinary care of the animals; and Drs. Fumiko Seki and Junichi Hata for obtaining the CT images. This work was financially supported by the Brain/MINDS (Brain Mapping by Integrated Neurotechnologies for Disease Studies) project from the Japan Agency for Medical Research and Development (JP18dm0207001, JP19dm0207001, and JP20dm0207001 to H.O.; JP20dm0207069 to M.K.); by JSPS KAKENHI (JP19H04993 to M.K.; 19K20653 to T.K.), and by internal budgets from Keio University including the Program for the Advancement of Research in Core Projects on Longevity of the Keio University Global Research Institute from Keio University (to H.O.).

## Author contributions

T.K and H.O. conceived the project. M.K., N.I., and T.Y. provided ECoG methodology. T.K developed the behavioral system. T.K and M.K. performed the surgery and the recording. T.K. analyzed the data. All authors discussed the results, and contributed to shape the final intellectual product. T.K wrote the manuscript with feedback from M.K., N.I., T.Y., and H.O.

## Competing interests

The authors declare no competing interests.
