## [Transparent Peer Review File · Communications Biology]

Reviewers' comments:

Reviewer #1 (Remarks to the Author):

In this manuscript, the authors present ECoG recordings from awake, behaving marmosets viewing natural movies. They analyze peri-saccadic activity patterns to understand the temporal dynamics among areas in visual cortex, after first performing a careful analysis concerning afferent and efferent contributions to peri-saccadic activity patterns.

The results reported in this manuscript are novel and interesting: instead of activity progressing either top-down, from higher to lower visual regions, or bottom-up, from lower to higher regions, the HG signal begins in MT, progresses through the dorsal stream to early areas, and then travels through the ventral stream. If true, this would be a novel pattern of behavior-related activity in the visual system, demonstrating that sophisticated spatiotemporal patterns of activity may be present in the visual system.

Several important points need to be addressed. The authors may want to distinguish more carefully between "Granger causation" (studied here) and "information", which are two similar but distinct notions. The authors should also relate their results to other studies that have found traveling waves of activity (particularly whole-brain rotating waves, which have previously been observed only during sleep).

Major comments:

Figure 2b: The computation of t-values is good, but still difficult to interpret based on the baseline activity pattern. It would be helpful to have a detailed comparison between the reported measure and the raw HG signal. Further, the Methods states that the baseline was calculated 1200-900 ms before and 900-1200 ms after the saccade. Given the previous statement that saccades happen frequently (every ~250 ms), is there not a potential confound with peri-saccadic activity being included in the baseline?

Figure 2c: Time here is from saccade onset. Is that the case through the rest of the paper? This was difficult to find in the main text or Methods.

Figure 4a: Is this an event-averaged representation?

Figure 4d: The computation of the power spectrum here should be much more clear, and the units should be checked.

Figure 6: There is a typo in the caption before panel (d).

Line 239: "Rosa and Tweedale claimed that the MT complex can be thought of as an additional primary visual area as it has direct input from the lateral geniculate nucleus": The motivation for this citation is not clear. Is the input from LGN to MT unique in the visual system except for V1? If this is the case, please make this clear; otherwise, the sentence is difficult to understand.

Reviewer #2 (Remarks to the Author):

This paper describes the flow of activity across cortical areas during natural visual exploration. The authors collected and analyzed electrocorticographic recordings from electrodes covering most of the lateral surface of four marmoset monkeys while they visually explored movies. Their main finding was that the dorsal and ventral streams were activated one after another by a traveling signal in sequence following saccades, and specific areas of the brain were dominated by efferent or afferent signals.

They also demonstrated that the timing of saccades can occur at varying time points while the pattern traverses the brain, implying that motor commands and visual processing are occurring parallelly.

In general, the characterization of the flow of high frequency signals from dorsal, through V1, to ventral streams during visual exploration is both novel and important. However, in its current form, the paper lacks clarity in the computation of efferent and afferent patterns of information and how the timing of activity implies parallel computation for saccade generation. Thus, overall, it is difficult to connect the paper's central claim about spatiotemporal dynamics of traveling activity to the regional specificity of efferent and afferent patterns and parallel computation.

Major comments:

* In analyzing the differences in activity between different fixation durations, there appears to be a relationship between fixation duration and the power of subsequent peaks that is not explored (Figure 6A). The authors should demonstrate that this is not problematic for their claims.

* I suggest expanding the data analyses to show results from individual animals. In Figure 6B, the point that activity patterns are drastically different at the end of different fixation durations is not clear when averaged across animals. Patterns described in lines 205-211 are not clear in Figure 6B. Perhaps showing the activity profiles of individual animals may demonstrate this difference more clearly. Focusing on a specific animal may also clarify regional differences for different fixation durations that are difficult to distinguish in Figure 6B when averaged across animals. [a]

* In Figure 3C-E, the methods used to compute and identify the differences between the Efferent - Afferent signals is not clearly discussed. For instance, since efferent and afferent signals include both contralateral and ipsilateral patterns in different regions, are contralateral and ipsilateral signals separated when computing bias and Efferent-Afferent differences? [b]Relatedly, the explanation of how bias scores are computed is not clear (Lines 434-438).

* In general, the paper would be greatly strengthened if the authors use more unambiguous, specific, and consistent terminology throughout.[c] For instance, the traveling pattern is described as a traveling signal, traveling gamma wave, and as a discrete traveling packet of information. Additionally, terms such as "cortical information dynamics" (40, 70), "cortical energy dynamics" (198), and "information flow" (48, 93, 232) are difficult to parse. The terms "signal flow" and "flow of information" have different implications.

Minor comments:

* Figure 2D does not exhibit a clear relationship between power and latency. This plot is confusing. However, if there is a significant difference in power between regions, the authors should alter the plot or analysis to better explain such a relation.

* The authors should clarify the interpretation of efferent and afferent information with a discussion of why the maximal timing of bias for both efferent and afferent are 0-70ms and not when the corresponding areas are activated by the traveling signal as shown in Figures 2 and 4. [d]In particular, the authors may clarify their interpretation of the timing of efferent and afferent signals in lines 132-141.

* The text claims that the traveling signal is an "information packet". This interpretation that information is being transmitted from one region to another is not evident given that the pattern is a traveling change in high frequency power rather than a traveling oscillation. The specific finding here that is demonstrated concerns patterns of gamma activity. I would write the paper in a way that complies more closely with that result, and then make it more clear that any claims regarding 'information' are speculative.

* In Figure 2, time axes for A and B should line up for visual interpretation of aligned signals.

- * The second Figure 2D should be changed to E, and the order of regions in the first panel d should be by latency.
- * Figure 4d can be plotted as a histogram of counts rather than power.
- * Figure 5a requires axis labels.

Other thoughts not for review:

- * Why high gamma signal? Are there traveling signals of beta saccade related activity
 - * Seems to be establishing points about more motor/efferent signals in dorsal/occ and vlpc, which is established in lit (specific papers)
 - * do saccades every 200ms depend on changes in moving pictures they see? Would differently timed movies elicit different timing of saccades
 - * -should discuss relevance of frequency of saccade? Is animal prone to saccades every 200ms because of duration of traveling wave?
- [a]i found this a bit hard to understand. could you be a little more specific in terms of what you are suggesting?
- [b]can you be more specific about what exactly is unclear here?
- [c]can you give examples to help them?
- [d]can you be more specific about this? be specific about the page/line number.

Reviewer #3 (Remarks to the Author):

Summary

Using marmoset monkeys as animal model, Kaneko et al. offer some interesting insights on the macroscopic high gamma power dynamics during free visual exploration. Their results show HGP increases in the dorsal stream preceding V1 activation and followed by ventral stream activation. Such HGP increases are detected every 200 ms. The paper title suggests that there is a causal relationship

between this rhythmic pattern and marmosets' saccades, but no evidence is provided to support this behavioral correlation.

The paper includes some attention-catching claims about 'information packaging' and 'travelling waves' in the context of oculomotor behavior. However, the presented data and analyses fall short in substantiating those claims. Moreover, the authors should verify that some of the paper results hold true independently of some methodological choices they took without offering explicit justification (see below).

General & Conceptual issues

- The most highly pitched results in the paper are derived from a statistical surrogate signal and not from the high gamma power data itself (see details below). As it was calculated, this surrogate signal incorporates the alignment of the data to saccade onset for baseline normalization. This fact seems to be completely ignored by the authors when interpreting their own results and can critically affect their conclusions in the 'parallel computation' section (see details below).
- Critically, the analyses presented in the paper only take into account increases in the high gamma power signal (see below). Given the growing evidence on the functional role of neural suppression in oculomotor behavior at different levels of the visual pathway, the decision to exclude any mention or analysis of neural suppression demands further explanation.
- The Methods section is incomplete and unspecific. It is crucial that for each analysis the authors explicitly state what was the signal used as input (i.e. the high gamma power time series, its t-value statistical surrogate or the LFP signal).
- Neural recordings were collected during a free-viewing task and a 'passive task'. In the text, it's not clear what was the hypothesis that demanded a comparison between these 2 conditions. Results from the passive task are only superficially and sporadically mentioned, and Figures are relegated to the Supplementary Section. In the discussion, please elaborate more on how neural dynamics differ between these 2 visual conditions.
- When describing their results, Kaneko et al. use terminology that is not justified by the specific analysis they performed. For example, they use the term 'information package' without using any information theoretical measure when describing the results presented in Figure 4. Moreover, they repeatedly refer to 'travelling waves' while their analyses are constrained to the power of the high gamma frequency band. In contrast, 'travelling-wave' analyses generally involve the quantification of phase differences across recording sites (and usually focus on low frequency oscillations).
- This liberal use of the terms 'information package' and 'travelling waves' compromises the conceptual soundness of the paper. For example, an explanation of how their discrete account of oculomotor information fits into the travelling waves framework is completely absent. Relatedly and more generally, some of the ideas put forward as part of the results are more appropriate for an opinion paper and would still require argumentative substantiation. For example, the authors claim that "visual information does not flow continuously, but is rather like a discrete traveling packet in natural visual behavior".
- While the data and analysis presented in this paper are valuable, I encourage the authors to describe their results using more concise and descriptive language. Please avoid terms as 'cortical energy dynamics' (line 198) or 'signal packet' (line 204).
- Furthermore, the paper would benefit from grammatical and stylistic review to improve clarity and readability. Some sentences are difficult to understand, and the argumentative strategy of the authors is hard to follow.

- Finally, the significance of the paper and the reach of its results would greatly benefit from additional analysis of neural activity in the low frequency oscillations. In contrast to high frequency oscillations, classically considered of local origin, low frequency oscillations are thought to carry information over long distances. Therefore, these oscillations would offer a more parsimonious framework for the authors hypothesis and interpretations. For example, the intriguing signatures of a 5Hz rhythmicity in their results (Fig4) could be possibly explained by the coupling of high gamma power to theta frequency oscillations.

Major comments:

In line 119: Efferent and Afferent signals: The authors aimed to quantify the degree to which the ECoG signal is biased either to efferent or afferent processing. They later claimed that: "if the signal is motor-related, then the saccade-evoked activity is larger for a contralateral saccade than for an ipsilateral saccade (Bash et al., 1991)." The authors measured responses to the direction of the saccades (ipsilateral or contralateral) and from there conclude if in a determined cortical area the signal is "motor" or "sensory related." Moreover, they assert: "In this way, the afferent information should result in larger neural activity for ipsilateral saccades than for contralateral saccades." The authors seem to be basing this analysis and conclusions in two critical papers (Bash et al., 1991 and Chen et al., 2016). However, I think several things need clarification and are important to the conclusions that derive from this section:

- Bash et al., do show cells with directional modulation by saccades in the inferior parietal lobule (IPL). These authors indeed found a prominent contralateral modulation (69% of cells). However, this depended on the evaluation period (e.g. pre-saccadic, post-saccadic), type of modulation (e.g. excitatory or inhibitory) and can be variable ("around the time of the saccade, this trend reversed, and most inhibitory responses showed ipsilateral preferred directions"). To my understanding paper does not argue that a contralateral activation mainly speaks of an "efferent signal." Notably, they do express that the directionality of the modulation seems to be related to a motor and not a sensory signal: "Cells fire for saccades in their preferred directions regardless of whether a visual stimulus has or has not fallen in their receptive fields," but this is independent of the specific preferred direction of the cells.

- Chen et al., 2016 show evidence that inactivation of the ventral part of LIP did decrease the expression of contralateral saccades. They show that the dorsal section of LIP shows no directional modulation while the ventral lip is widely related to motor signals. However, it needs to be noted that the area considered to process sensory information had no directional preference and not an ipsilateral preference. A directional modulation in this case, even if it is ipsilateral would be considered as a motor signal as it would process or even "plan" for the direction of the movement.

- The present paper is challenging to place apart from the difference between visual or "afferent" signals to motor or "efferent signals." I cannot agree that the ipsilateral dominance of the MT has to be interpreted as being of an afferent signal. Actually, the timing of activation (early after saccade onset) and the directional modulation could be evidence of the contrary. In fact, cellular preference of directionality has been considered as an essential criterion of a corollary discharge (e.g. Crapse and Sommer, 2008).

To help dissect out the relative contributions of the afferent and efferent components did the authors consider performing the experiments in the dark? It is known that efference copies are maintained in the dark condition which would allow attribution of specific components of what is observed to the efference copy without visual input.

- While the analyses included in the paper are constrained to the high gamma band, in the conclusions and discussion the results are inappropriately generalized to claims about the LFP signal or neural activity in general. High frequency and low frequency oscillations are known to encode complementary information (REFS). For example, the discrete nature of high gamma increases in the visual areas does not rule out the possibility that low frequency oscillations provide a continuous flow of

information.

- As mentioned before, in the present paper low frequency oscillations are excluded from the analyses and the discussion, even when 'travelling-waves' are persistently mentioned. A comprehensive account of oculomotor behavior demands exploring the presented results in the context of low frequency oscillations. I strongly encourage the authors to look at the low frequency domain and to include phase analyses using the formal 'travelling-wave' framework (see for example Muller et al. 2018).

- o Both in resting state and after visual stimulation, low frequency travelling waves in the 5-10 Hz frequency range have been shown to move from posterior to anterior regions of the brain. These findings could partially account for the results here presented without specific contributions of oculomotor behavior.

Results & Methods

- My main concern with this paper is that most of the presented results (including the 'travelling wave' and 'parallel processing' analyses) do not rely on the high gamma power series, but on a statistical surrogate. Crucially, that surrogate is obtained by comparison to an arbitrary baseline period, defined with respect to saccade onset. Critically, this data (saccade-onset aligned and baseline-normalized) is used for all following analysis. This point is mentioned in the methods but seems to be forgotten when interpreting the results in the 'travelling wave' and 'parallel processing' sections.

- o The authors explain that "In each frequency range, the signal modulation of the perisaccadic period was measured as t-statistics of each time point by comparison to baseline activity, which were median time points far after or before the saccade (1200 to 900 ms before saccade and 900 to 1200 ms after saccade)". They indeed "obtained a single time series of the high-gamma signals as the average of t-statistics across 100–160 Hz"

- o They justified this choice appealing to 1/f power-frequency relationship: "We analyzed this t-value time series data instead of the physical power of the raw ECoG signal because the high-frequency signal has small power in nature, but it still contains meaningful information in terms of the signal to noise ratio; thus, the difference in net signal power in each frequency range can be canceled out by computing the t-statistics, taking into account the signal to noise ratio in each frequency range".

- o This justification is inadequate as many forms of baseline normalization applied on power data per frequency (sub)band can solve this issue (z-score, dB).

- o I do not understand why after going through the computational effort to run these statistics the authors used the resulting t-values but completely ignored the p-values from the test. Usually the 'statistical approach' is used to make use of only those time points and/or those channels that satisfy some threshold significance level (based on the p-values).

- In any case, the most problematic effect on some of their results comes from the fact that the used baseline is arbitrarily defined (between +/-900 and +/-1200 ms) with respect to saccade onset.

- Notice that saccades happen approximately every 200 ms in average; therefore, such distant baseline may not be 'silent' (as likely there are other saccades in the middle).

- It's also problematic that the authors ran a two-tailed t-test to obtain this surrogate signal (obtaining both positive and negative t-values) and then excluded negative t-values for further analyses without offering any justification.

- o Given the extensive evidence on the role of neural suppression in oculomotor behavior (REFS!), the decision to exclude negative t-values from their analyses of neural dynamics (at least for those shown in Figures 5 and 6) requires explanation.

- o If the authors decided to exclude negative t-values (already a controversial choice), they could have used in principle a one-tail t-test to obtain the statistical surrogate.

- o In the end, those calculated negative t-values are only displayed in the 'statistical' spectrograms shown in Figures 2 and 3 (which make them look similar to common power spectrograms).

- o Correspondingly, their latency analyses seem to be identifying only positive peaks and ignoring suppression peaks.

o I would strongly recommend that instead of using the specified arbitrary period as baseline the authors compare perisaccadic time points in the actual data with a surrogate distribution obtained from randomly sampling the HGP time series. This procedure can be performed in the context of a permutation test and represents a robust approach that does not involve the selection of a specific and arbitrary baseline period. Alternatively, the authors could z-score the whole time-series corresponding to the recorded session after artifact rejection.

- Signal Flow under natural visual behavior

o The paper lacks a description of how latencies represented in Figs. 2 d,e were calculated. Was the peak defined by the local maximum? Were peak magnitudes and latencies detected per saccade and then averaged per site, or identified in the average over saccades per site? Please add this description to the methods section.

- Distinct patterns of motor- and visual-related signals

o Several previous studies have functionally described single neurons as visual, motor or visuomotor in the context of oculomotor behavior. In the same line, the authors aim to quantify the contribution of visual and motor signals to the recorded LFP. For that purpose, the authors try to borrow a strategy from one previous single-cell study in which visual stimulus location was controlled in the context of two structured paradigms (Barash et al. 1991, using a double-saccade and back-saccade paradigm). This strategy consists in the distinction between motor and visual signals according to the comparative response magnitude to ipsi- and contralateral saccades (in relation to the brain hemisphere being recorded). In that single cell study, Barash et al. not only precisely control the location of the visual target, but also knows the receptive field and directional tuning of the single neurons they are studying in the lateral intra-parietal area (LIP).

o In the present study by Kaneko et al. there is not a clear description of how visual stimuli location was controlled and it's unclear to me how their argument works in the absence of that information. The scheme in Figure 3a could be misleading by representing only one of the many possible cases. In a free visual exploration paradigm, depending on its starting point and its magnitude, and ipsilateral or contralateral saccade can bring certain part of the scene into or outside the (recorded) hemisphere receptive field.

o Moreover, the authors' arguments are hard to follow and problematic especially with respect to the afferent signal. For example, the authors assert that "afferent information should result in larger neural activity for ipsilateral saccades than for contralateral saccades". Again, this does not hold true in their paradigm.

o Moreover, Kaneko's argument does not take into account that different areas of the visual pathway are known to have differently sized 'receptive fields' (retinotopic selectivity).

o These considerations could explain why the authors did not find the expected afferent or efferent patterns in some areas of interest, and demands more careful interpretation of the data.

o Instead of relying on the conceptual scheme shown in Fig3a (appropriate only for more controlled paradigms at another level of description), the authors could compare activity patterns elicited by 2 possible LFP data alignments: saccade onset and fixation onset. This is the preferred approach in many published studies at the meso and macroscopic level.

- Spatiotemporal dynamics:

o According to the authors description, high-gamma spatiotemporal dynamics differed between the free viewing and the passive condition. The latter, only shown in S1F, exhibits the classically described pattern from occipital cortex to the dorsal and ventral streams. How do the authors precisely explain this difference?

Discussion

- After using the 'traveling wave' terminology repeatedly, the authors fail to include any reference to seminal work in the field. A growing number of papers has described the spatiotemporal dynamics of travelling-waves in humans and non-human primates. The literature on the functional role of travelling-waves elicited by visual stimulation is especially relevant in this case (REFS).

Figures

- Figure 2:
 - o The low frequency cut (y axis) in the statistical spectrograms shown in panel a, seems to be 50 Hz. Please add a label to the low limit, so it's clear activity below 50Hz is not shown.
 - o Two panels are labelled as (d).
- Figure 3:
 - o It would be beneficial to increase the size of the spectrograms in panel b. The spectrograms for ipsilateral and contralateral saccades are difficult to see and seem to be important for the distinction between afferent and efferent signals made by the authors.
 - ♣ Particularly, I'm intrigued by the apparent acute transition seen at time 0, even for 'background activity'. This is particularly obvious for MT panels (ipsilateral and contralateral saccades). Where each side of the spectrograms normalized to their one sided baseline?
 - ♣ The baseline comparison period is included in these plots (-1200 to 900 ms and 900 to 1200 ms) and should correspond to zero, but that does not seem to be the case in the shown color scale.
- Figure 4:
 - o The results shown in Figure 4 are definitely interesting, but the corresponding methods are not described. How was the center of gravity of the high gamma signal calculated? Please describe in the Methods section.
 - o Please show individual subject figures as Fig4B in the supplementary Information. This is important to verify that the general/whole trajectory dynamics are preserved at the single subject level.
 - o As mentioned before, the data patterns observed in Figure 4 demand a phase-amplitude coupling analysis to test the hypothesis that high gamma power is coupled to the phase of theta oscillations (5 Hz). Further analysis would be required to make the point that those theta oscillations are indeed 'travelling waves' but given the extensive electrode coverage the authors are in the privileged position of performing those analyses.
- Figure 5:
 - o A1 dynamics are shown as control. Including a site out of the visual pathway adds value to the results, but this control is surprisingly not mentioned or described in the main text. Do other sites far from the visual pathway trajectory look similar (flat)?
- Figure 6:
 - o Were all recording sites included in the analysis corresponding to Figure 6a? Please provide a rationale for including sites out of the visual pathway.

We would like to thank the editor and three reviewers for their constructive comments, which helped us to improve our manuscript. Below, we address all of their comments in a point-by-point manner and discuss the subsequent revisions.

The original comments from the reviewers are shown in black font, and our replies are shown in blue font.

Reviewers' comments:

Reviewer #1 (Remarks to the Author):

In this manuscript, the authors present ECoG recordings from awake, behaving marmosets viewing natural movies. They analyze peri-saccadic activity patterns to understand the temporal dynamics among areas in visual cortex, after first performing a careful analysis concerning afferent and efferent contributions to peri-saccadic activity patterns.

The results reported in this manuscript are novel and interesting: instead of activity progressing either top-down, from higher to lower visual regions, or bottom-up, from lower to higher regions, the HG signal begins in MT, progresses through the dorsal stream to early areas, and then travels through the ventral stream. If true, this would be a novel pattern of behavior-related activity in the visual system, demonstrating that sophisticated spatiotemporal patterns of activity may be present in the visual system.

Several important points need to be addressed. The authors may want to distinguish more carefully between "Granger causation" (studied here) and "information", which are two similar but distinct notions.

When we refer to the results of Granger causality analysis, we have replaced "information" with more appropriate wording.

Our revision appears on Line 167 in the main text,

Granger causality analysis also supported the presence of signal influence from the dorsal visual areas to the posterior part of the occipital visual areas (Supplementary Fig. 6).

And also in Supplementary figure 6,

The signal in the MT complex and PPC influenced the dorsal occipital regions, and the dorsal occipital regions also influenced the earlier visual areas.

The authors should also relate their results to other studies that have found traveling waves of activity (particularly whole-brain rotating waves, which have previously been observed only during sleep).

We are grateful to the reviewer for drawing our attention to the analysis of travelling waves with low frequency oscillations. A recent report of multi-unit recording in marmosets showed that spontaneous travelling waves of theta band oscillations can be detected in awake animals, and their phase is predictive for the sensitivity of detecting a visual stimulus. In the original draft, we focused on the transition of the high-gamma activity peak, and now we added analysis of the phase of the theta band, according to previous studies of travelling waves. We indeed found that the travelling wave had a similar pattern as the high-gamma signal. We also found phase-amplitude coupling between theta and high-gamma activity, suggesting that the transient visual response driven by saccade was transmitted from the dorsal to ventral stream and is accompanied by a travelling wave of theta phase oscillations.

This revision appears in Fig. 6 d-i, and lines 169-186.

Major comments:

Figure 2b: The computation of t -values is good, but still difficult to interpret based on the baseline activity pattern. It would be helpful to have a detailed comparison between the reported measure and the raw HG signal. Further, the Methods states that the baseline was calculated 1200-900 ms before and 900-1200 ms after the saccade. Given the previous statement that saccades happen frequently (every ~250 ms), is there not a potential confound with peri-saccadic activity being included in the baseline?

Instead of reporting only t -values using an arbitrary time window as the baseline, we simply used the z -score of each frequency band. We further computed p -values based on a randomization test in which peri-saccade activity generated by pseudo-saccade onset was randomly generated. The results were consistent with our original draft.

This change appears on Figure 2b and all other analysis. See also lines 430-439 in the methods section.

Figure 2c: Time here is from saccade onset. Is that the case through the rest of the paper? This was difficult to find in the main text or Methods.

It was from saccade onset throughout the manuscript, except when it was explicitly mentioned as being from offset. We changed the Methods sections and the figure legend to describe this clearly (line 437 and 783 in the main text).

Figure 4a: Is this an event-averaged representation?

Yes. This shows the average activity pattern from all subjects. It was not possible to capture this pattern in a trial-by-trial manner as the signal was subtle. We now state this explicitly in the Methods section (line 476).

Figure 4d: The computation of the power spectrum here should be much more clear, and the units should be checked.

According to the substantial revision of the figures and additional analyses performed, we removed this panel.

Figure 6: There is a typo in the caption before panel (d).

We corrected this in the revised manuscript.

Line 239: “Rosa and Tweedale claimed that the MT complex can be thought of as an additional primary visual area as it has direct input from the lateral geniculate nucleus”: The motivation for this citation is not clear. Is the input from LGN to MT unique in the visual system except for V1? If this is the case, please make this clear; otherwise, the sentence is difficult to understand.

We revised the manuscript as suggested by the reviewer.

This appears on page 12, lines 264-267

Rosa and Tweedale claimed that the MT can be thought of as “an additional primary visual area” as it has direct input from the lateral geniculate nucleus. In this sense, V1 is not a unique cortical area that receives the earliest visual input from subcortical

structures.

Reviewer #2 (Remarks to the Author):

This paper describes the flow of activity across cortical areas during natural visual exploration. The authors collected and analyzed electrocorticographic recordings from electrodes covering most of the lateral surface of four marmoset monkeys while they visually explored movies. Their main finding was that the dorsal and ventral streams were activated one after another by a traveling signal in sequence following saccades, and specific areas of the brain were dominated by efferent or afferent signals. They also demonstrated that the timing of saccades can occur at varying time points while the pattern traverses the brain, implying that motor commands and visual processing are occurring parallelly.

In general, the characterization of the flow of high frequency signals from dorsal, through V1, to ventral streams during visual exploration is both novel and important. However, in its current form, the paper lacks clarity in the computation of efferent and afferent patterns of information and how the timing of activity implies parallel computation for saccade generation. Thus, overall, it is difficult to connect the paper's central claim about spatiotemporal dynamics of traveling activity to the regional specificity of efferent and afferent patterns and parallel computation.

Major comments:

* In analyzing the differences in activity between different fixation durations, there appears to be a relationship between fixation duration and the power of subsequent peaks that is not explored (Figure 6A). The authors should demonstrate that this is not problematic for their claims.

We are not sure for the precise reason why the power of subsequent peaks differed across fixation duration. The signal pattern after fixation initiation (0–200 ms; 0–130 ms for the 150 ms fixation condition) was quite similar regardless of how long the subsequent fixation was maintained, however, the activity profiles just before the next saccade differed drastically according to the prior fixation duration. This difference might explain the difference in the power of the subsequent peak. In this sense, we believe that the difference in the power of the subsequent peak is consistent with our main claim and is not problematic in terms of our main points in this figure.

* I suggest expanding the data analyses to show results from individual animals. In Figure 6B, the point that activity patterns are drastically different at the end of different fixation durations is not clear when averaged across animals. Patterns described in lines 205-211 are not clear in Figure 6B. Perhaps showing the activity profiles of individual animals may demonstrate this difference more clearly. Focusing on a specific animal may also clarify regional differences for different fixation durations that are difficult to distinguish in Figure 6B when averaged across animals. [a]

We agree that, in the original draft, it was difficult to see the difference in activity pattern at fixation offset. Here, we reorganized the figure so that the activity at fixation offset for different fixation durations was presented in larger panels. In addition, activity power was normalized for each condition only for visualization purposes, as the power is relatively small for longer fixation periods (and difficult to see). We also added individual results in Supplementary Fig. 10. This modification appears in Figure 8 and Supplementary Figure 10.

* In Figure 3C-E, the methods used to compute and identify the differences between the Efferent - Afferent signals is not clearly discussed. For instance, since efferent and afferent signals include both contralateral and ipsilateral patterns in different regions, are contralateral and ipsilateral signals separated when computing bias and Efferent-Afferent differences? [b]Relatedly, the explanation of how bias scores are computed is not clear (Lines 434-438).

In the original draft, we simply computed the difference of activity amplitude of contralateral and ipsilateral saccades in an attempt to dissociate the efferent and afferent signals. However, as Reviewer #3 argues, this analysis is not sufficiently convincing to clearly separate efferent and afferent information solely based on the biased activity toward ipsilateral or contralateral saccades. In the revised manuscript, we mentioned that, according to this analysis, the VLPFC has a large degree of contralateral dominance at saccade onset and indicated that the processing of this region put a large emphasis on target selection or saccade generation. We believe this interpretation is straightforward from the analysis we performed here. In addition, the Methods section was revised to clarify how we computed the ipsi/contralateral biases. This change is on ...

* In general, the paper would be greatly strengthened if the authors use more unambiguous, specific, and consistent terminology throughout.[c] For instance, the traveling pattern is described as a traveling signal, traveling gamma wave, and as a discrete traveling packet of information. Additionally, terms such as “cortical information dynamics” (40, 70), “cortical energy dynamics” (198), and “information flow” (48, 93, 232) are difficult to parse. The terms “signal flow” and “flow of information” have different implications.

We substantially revised the text to improve readability, especially the parts in which we used the terms “travelling wave,” “cortical energy dynamics,” and “signal packet” etc. as “information flow.” We believe the unnecessary speculative or augmentative descriptions have been removed entirely from the revised version.

“Travelling pattern” is now described as “trajectory of high gamma signal”(e.g., line 156-168). Please note that we still used “travelling wave” in the analysis of low frequency oscillation which we added in the revised manuscript. This analysis is consistent with what previous studies did to describe the travelling wave. “Cortical information dynamics” is now “cortical neural dynamics”(e.g., line 66). “Cortical energy dynamics” is now “the total activity of the whole brain”(line 198). “Flow of information” is now “signal flow” or “neural dynamics”(line 92, 258).

Minor comments:

* Figure 2D does not exhibit a clear relationship between power and latency. This plot is confusing. However, if there is a significant difference in power between regions, the authors should alter the plot or analysis to better explain such a relation.

We agree that the panel was not relevant to the logical flow of the figure, so we have removed it.

* The authors should clarify the interpretation of efferent and afferent information with a discussion of why the maximal timing of bias for both efferent and afferent are 0-70ms and not when the corresponding areas are activated by the traveling signal as shown in Figures 2 and 4. [d]In particular, the authors may clarify their interpretation of the timing of efferent and afferent signals in lines 132-141.

As described in a previous comment, we limit our interpretation in terms of contralateral dominance (if it exists), and not a distinction of efferent and afferent processing. In this revision, the original description corresponding to this comment was removed entirely.

The original text on lines 132–141 was as below:

Similarly, the posterior parietal region also showed a signal dominated by efferent information (Fig. 3b–f). It might appear slightly peculiar that the efferent/motor-related signal reached its peak after saccade onset and was sustained for a while after saccade offset; however, similar properties have been reported using single-unit recording³⁵ or the simultaneous recording of single units and local field potentials from the same contact³⁹, which is probably due to the activity of visuomotor neurons.

Interestingly, the MT complex, with the earliest cortical activity, showed the opposite trend to the efferent signals (Fig. 2b, d, f), where ipsilateral saccades showed larger responses, indicating that the signal in this region is likely dominated by afferent information. Such a very rapid afferent response after saccades corresponds to the findings of a single-unit recording and neuroanatomical study of the macaque MT complex⁴⁰.

The answer to the reviewer's comments is not simple. We assume this is because the lateral bias appeared differently across different types of neurons (visual, motor, and visuo-motor neurons, saccade suppression type/non-suppression type, etc.). The receptive field and activation timing should differ across those neuron types. Perhaps, population activity would be large with neurons showing a weaker lateral bias, and activity latency would be short if neurons with short latency had a stronger lateral bias. This is highly speculative and difficult to connect with the actual neuronal types that have been reported in the literature. We had the impression that it would be worthwhile to simulate ECoG signals from the combination of known neuron types, but this is beyond the scope of the present study. Therefore, in the revised manuscript, we moved the argument about afferent and efferent distinction from the Results section to the Discussion section and indicated that this was an interpretation, and not a conclusion. This argument appeared in lines 281 – 304.

* The text claims that the traveling signal is an “information packet”. This interpretation that information is being transmitted from one region to another is not evident given that the pattern is a traveling change in high frequency power rather than a traveling oscillation. The specific finding here that is demonstrated concerns patterns of gamma activity. I would write the paper in a way that complies more closely with that result, and then make it more clear that any claims regarding ‘information’ are speculative.

In the revised manuscript, first, we removed the term “information packet,” which could not be concluded from the analysis we performed. Second, we added analysis of the travelling wave with phase shift at the lower frequency range. Interestingly, the pattern of wave propagation was consistent with the trajectory of the high-gamma signal, and we also observed coupling of the low frequency phase and high-gamma amplitude around saccades. These results indicate that the high-gamma signal may be carried on such low frequency oscillations. These results appear on lines 169-186 and fig.4 d-i.

* In Figure 2, time axes for A and B should line up for visual interpretation of aligned signals. We revised Figure 2 so that the x-axes were comparable across both panels.

* The second Figure 2D should be changed to E, and the order of regions in the first panel d

should be by latency.

We revised the organization of Figure 2.

* Figure 4d can be plotted as a histogram of counts rather than power.

According to the substantial revision of the figures and additional analyses, we have removed this panel.

* Figure 5a requires axis labels.

We have added labels to the axis labels(This is Fig7a in the revised manuscript.).

Other thoughts not for review:

* Why high gamma signal? Are there traveling signals of beta saccade related activity

Yes, we have reported the additional results regarding this point.

* Seems to be establishing points about more motor/efferent signals in dorsal/occ and vlpfc, which is established in lit (specific papers).

We referred to those studies in the Results and Discussion sections, while the connection between spike recording and ECoG is still difficult to establish.

* do saccades every 200ms depend on changes in moving pictures they see? Would differently timed movies elicit different timing of saccades

We are very interested in what visual features trigger different types of saccades and the differences in neural architecture involved in different types of saccades. This will be the focus of our next study.

* -should discuss relevance of frequency of saccade? Is animal prone to saccades every 200ms because of duration of traveling wave?

We argued this in the Discussion section(lines 314-321).

[a]i found this a bit hard to undrstnad. could you be a littel more specific in terms of what you are suggesting?

[b]can you be more specific about what exactly is unclear here?

[c]can you give examples to help them?

[d]can you be more specific abou this? be specific abou the page/line number.

Reviewer #3 (Remarks to the Author):

Summary

Using marmoset monkeys as animal model, Kaneko et al. offer some interesting insights on the macroscopic high gamma power dynamics during free visual exploration. Their results show HGP increases in the dorsal stream preceding V1 activation and followed by ventral stream activation. Such HGP increases are detected every 200 ms. The paper title suggests that there is a causal relationship between this rhythmic pattern and marmosets' saccades, but no evidence is provided to support this behavioral correlation.

We changed the title from "*Cortical neural dynamics that generate the rhythm of natural visual behavior*" to "*Cortical neural dynamics underlying the rhythm of natural visual behavior*". In the manuscript, we haven't addressed the causal relationship between neural signals and the

rhythm of saccades. However, we believe our results elucidated the meaningful relationship between them.

The paper includes some attention-catching claims about ‘information packaging’ and ‘travelling waves’ in the context of oculomotor behavior. However, the presented data and analyses fall short in substantiating those claims. Moreover, the authors should verify that some of the paper results hold true independently of some methodological choices they took without offering explicit justification (see below).

General & Conceptual issues

- The most highly pitched results in the paper are derived from a statistical surrogate signal and not from the high gamma power data itself (see details below). As it was calculated, this surrogate signal incorporates the alignment of the data to saccade onset for baseline normalization. This fact seems to be completely ignored by the authors when interpreting their own results and can critically affect their conclusions in the ‘parallel computation’ section (see details below).

As suggested by the following specific comments, we report the z -score in each frequency band, instead of reporting t -values computed using an arbitrary time window as the baseline throughout the manuscript including the “Parallel computation” sub-section. The results were consistent with our original draft. The obtained z -scores tended to be small; however, because of the substantial amount of data we generated, the results were very robust. We confirmed our findings by computing p -values using a randomization test in which peri-saccade activity was generated by pseudo-saccade onset derived from saccade onset taken from different sessions using the ECoG signals. The revised results appeared throughout the revised manuscript (especially at Fig.2, Fig.8 and Supplementary Fig. 1).

- Critically, the analyses presented in the paper only take into account increases in the high gamma power signal (see below). Given the growing evidence on the functional role of neural suppression in oculomotor behavior at different levels of the visual pathway, the decision to exclude any mention or analysis of neural suppression demands further explanation.

Thank you for this insightful suggestion. We analyzed the degree and timing of the suppression component. We found that such suppression can be seen along the entire ventral stream, and this was less prominent along the dorsal stream, except for the dorsal occipital regions. Interestingly, the timing of suppression differed drastically across the areas. The timing of suppression along the ventral stream was systematically changed according to the excitation timing of each area, so that the time difference between suppression and excitation remained consistent.

These results appear on lines 100-109 and Fig. 3.

- The Methods section is incomplete and unspecific. It is crucial that for each analysis the authors explicitly state what was the signal used as input (i.e. the high gamma power time series, its t -value statistical surrogate or the LFP signal).

In the revised manuscript, we explicitly mentioned whether the analysis was performed using the original time course or with surrogate statistics (such as the z -score).

These changes appear on lines 433-434, 444-445.

- Neural recordings were collected during a free-viewing task and a ‘passive task’. In the text,

it's is not clear what was the hypothesis that demanded a comparison between these 2 conditions. Results from the passive task are only superficially and sporadically mentioned, and Figures are relegated to the Supplementary Section. In the discussion, please elaborate more on how neural dynamics differ between these 2 visual conditions.

We put the passive-viewing task into Supplemental Information because we only performed this experiment with 2 of the 4 subjects. The motivation for comparing signal flow under these conditions was that the signal pattern of the whole brain during active visual exploration was quite different from that proposed in the so-called dual-stream hypothesis. Therefore, we hypothesized that the pattern we observed was specific to active visual exploration, and should not be seen when the stimuli changed passively. We included this argument in the Discussion section in the revised manuscript, and also please see the related reviewer's comment and our reply below. This change appears at lines 348-354.

- When describing their results, Kaneko et al. use terminology that is not justified by the specific analysis they performed. For example, they use the term 'information package' without using any information theoretical measure when describing the results presented in Figure 4. Moreover, they repeatedly refer to 'travelling waves' while their analyses are constrained to the power of the high gamma frequency band. In contrast, 'travelling-wave' analyses generally involve the quantification of phase differences across recording sites (and usually focus on low frequency oscillations).

In the revised manuscript, we rephrased these terms with more appropriate wording that concisely describes the results we observed. Specifically, we removed and rephrased "information packet," for example, "the peak of the high-gamma signal moved across regions." In terms of the "travelling wave," we analyzed the gradient of the phase in a lower frequency range. Please find the reply in the following specific reviewer's comment about the travelling wave. This revision lines 155-168, 169-186 and Fig.6.

- This liberal use of the terms 'information package' and 'travelling waves' compromises the conceptual soundness of the paper. For example, an explanation of how their discrete account of oculomotor information fits into the travelling waves framework is completely absent. Relatedly and more generally, some of the ideas put forward as part of the results are more appropriate for an opinion paper and would still require argumentative substantiation. For example, the authors claim that "visual information does not flow continuously, but is rather like a discrete traveling packet in natural visual behavior".

We completely removed those words/sentences that compromised the conceptual solidness of the original draft. Specifically, we removed "visual information does not flow continuously, but is rather like a discrete traveling packet in natural visual behavior" and the use of "information package." In terms of the travelling wave, we added phase analysis of a lower frequency range as used commonly in studies of travelling waves. This revision is mainly at lines 155-168.

- While the data and analysis presented in this paper are valuable, I encourage the authors to describe their results using more concise and descriptive language. Please avoid terms as 'cortical energy dynamics' (line 198) or 'signal packet' (line 204).

As suggested by the reviewer, we carefully revised the manuscript throughout to use more concise and descriptive language. Specifically, we rephrased "cortical energy dynamics" as, for example, "sum of the gamma signal across the cortex." We also removed the use of "signal

packet/information packet.” We simply mention that the high-gamma signal is spatiotemporally localized around saccades and the local peak moves across the cortex over time.

- Furthermore, the paper would benefit from grammatical and stylistic review to improve clarity and readability. Some sentences are difficult to understand, and the argumentative strategy of the authors is hard to follow.

We substantially revised the text to improve readability. We also sent our manuscript for grammatical correction and proof reading to a commercial editing service (NAI, Inc.). We believe unnecessary speculative or augmentative descriptions have been removed entirely from the revised version.

- Finally, the significance of the paper and the reach of its results would greatly benefit from additional analysis of neural activity in the low frequency oscillations. In contrast to high frequency oscillations, classically considered of local origin, low frequency oscillations are thought to carry information over long distances. Therefore, these oscillations would offer a more parsimonious framework for the authors hypothesis and interpretations. For example, the intriguing signatures of a 5Hz rhythmicity in their results (Fig4) could be possibly explained by the coupling of high gamma power to theta frequency oscillations.

Thank you for drawing our attention to phase analysis of lower frequency oscillations. We performed phase analysis and found that the phase of lower frequency oscillations formed a gradient across adjacent electrodes, and the directionality of phase shift had notable trends across the cortex. Interestingly, this pattern corresponded to the trajectory of the high-gamma activity peak. As phase-amplitude coupling was also observed, the high-gamma signal may be carried on the so-called travelling wave with low frequency oscillations.

The new analysis appeared in lines 169-186 and Fig.4 d-i.

Major comments:

In line 119: Efferent and Afferent signals: The authors aimed to quantify the degree to which the ECoG signal is biased either to efferent or afferent processing. They later claimed that: "if the signal is motor-related, then the saccade-evoked activity is larger for a contralateral saccade than for an ipsilateral saccade (Bash et al., 1991)." The authors measured responses to the direction of the saccades (ipsilateral or contralateral) and from there conclude if in a determined cortical area the signal is "motor" or "sensory related." Moreover, they assert: "In this way, the afferent information should result in larger neural activity for ipsilateral saccades than for contralateral saccades."

The authors seem to be basing this analysis and conclusions in two critical papers (Bash et al., 1991 and Chen et al., 2016). However, I think several things need clarification and are important to the conclusions that derive from this section:

- Bash et al., do show cells with directional modulation by saccades in the inferior parietal lobule (IPL). These authors indeed found a prominent contralateral modulation (69% of cells). However, this depended on the evaluation period (e.g. pre-saccadic, post-saccadic), type of modulation (e.g. excitatory or inhibitory) and can be variable ("around the time of the saccade, this trend reversed, and most inhibitory responses showed ipsilateral preferred directions"). To my understanding paper does not argue that a contralateral activation mainly speaks of an "efferent signal." Notably, they do express that the directionality of the modulation seems to be related to a motor and not a sensory signal: "Cells fire for saccades in their preferred

directions regardless of whether a visual stimulus has or has not fallen in their receptive fields," but this is independent of the specific preferred direction of the cells.

- Chen et al., 2016 show evidence that inactivation of the ventral part of LIP did decrease the expression of contralateral saccades. They show that the dorsal section of LIP shows no directional modulation while the ventral lip is widely related to motor signals. However, it needs to be noted that the area considered to process sensory information had no directional preference and not an ipsilateral preference. A directional modulation in this case, even if it is ipsilateral would be considered as a motor signal as it would process or even "plan" for the direction of the movement.

- The present paper is challenging to place apart from the difference between visual or "afferent" signals to motor of "efferent signals." I cannot agree that the ipsilateral dominance of the MT has to be interpreted as being of an afferent signal. Actually, the timing of activation (early after saccade onset) and the directional modulation could be evidence of the contrary. In fact, cellular preference of directionality has been considered as an essential criterion of a corollary discharge (e.g. Crapse and Sommer, 2008).

To help dissect out the relative contributions of the afferent and efferent components did the authors consider performing the experiments in the dark? It is known that efference copies are maintained in the dark condition which would allow attribution of specific components of what is observed to the efference copy without visual input.

We accept that it was not convincing enough to dissociate the efferent and afferent signals only from the biased activity on contra/ipsilateral saccades in the original manuscript. In fact, we tried to perform a saccade experiment under darkness before the initial submission; however, this proved to be technically challenging under our setup, and we were unable to obtain a reasonable amount of data. First, it was difficult to maintain awareness (and eye opening) of our animals under complete darkness after a few minutes of the experiment, even with sugar liquid rewards. Second, the eye morphology of marmosets makes eye tracking difficult to perform under darkness with our system, which is image-based detection of the contour of the pupil. The marmoset pupil is relatively large, and the exposed region of sclera is limited. Under complete darkness, as the pupil expanded, we were unable to precisely locate their direction of gaze.

In the revised manuscript, we only mentioned that, according to contra/ipsilateral saccade analysis, the VLPFC had a large degree of contralateral dominance at saccade onset and indicated that the processing of this region placed an emphasis on target selection or saccade generation. We believe the data and analysis support this interpretation. We removed the detailed argument on ipsilateral dominant activity as this may be derived from various factors such as receptive field size, corollary discharge, etc., as mentioned by the reviewer.

In addition, as suggested by the following comments, we added analysis of signal alignment by saccade onset or fixation onset. The results were quite intriguing. The excitation timing of the MT was better explained by saccade onset, but its suppression was determined by fixation offset (and this follows excitation, instead of in advance of saccade). On the contrary, suppression of the PPC, which occurred prior to excitation, was better explained by saccade onset, and the following excitation was better explained by fixation onset. In other words, the MT was active when the eyes were moving and suppressed when they were stationary, which was the opposite to what occurred in the PPC. The scope of this paper was to describe the neural dynamics of active vision across the cortex, so we believe it is worthwhile reporting how the activity dynamics differed across the regions that were activated prior to V1 (and the ventral stream) during natural visual behavior, even though we do not go into depth to reveal

the cellular level information representation of each dorsal area (which would be the focus of a future study).

The changes in response to this comment appear on lines 112-153, 280-304 and Fig. 4, 5 and Supplementary Fig. 2.

- While the analyses included in the paper are constrained to the high gamma band, in the conclusions and discussion the results are inappropriately generalized to claims about the LFP signal or neural activity in general. High frequency and low frequency oscillations are known to encode complementary information (REFS). For example, the discrete nature of high gamma increases in the visual areas does not rule out the possibility that low frequency oscillations provide a continuous flow of information.

As in the reply above, we included phase analysis of the lower frequency range, and found an interesting relationship between them. However, we agree that our results cannot be generalized to neural activity or local field potential. In fact, for example, the lower part of the gamma signal behaved slightly different from the high-gamma signal in our data and the boundary between the lower and higher gamma signals appeared systematically different among the ventral streams. However, these observations are beyond the scope of the current paper. As the high-gamma signal was shown to correlate with local neural activity in a previous study, we decided to focus on this frequency range in the present study. In the revised manuscript, we prepared described the limitation of the generalizability of the results.

This appears in the Discussion, lines 354-357.

- As mentioned before, in the present paper low frequency oscillations are excluded from the analyses and the discussion, even when ‘travelling-waves’ are persistently mentioned. A comprehensive account of oculomotor behavior demands exploring the presented results in the context of low frequency oscillations. I strongly encourage the authors to look at the low frequency domain and to include phase analyses using the formal ‘travelling-wave’ framework (see for example Muller et al. 2018).

- o Both in resting state and after visual stimulation, low frequency travelling waves in the 5-10 Hz frequency range have been shown to move from posterior to anterior regions of the brain. These findings could partially account for the results here presented without specific contributions of oculomotor behavior.

As in the reply to the previous comment, we included analysis on the travelling wave of the phase shift in the lower frequency range. Interestingly, the wave propagated from the posterior to anterior regions along the ventral stream, while its direction was opposite at the frontal, parietal, and dorsal occipital regions, and thus corresponds well with the trajectory of the high-gamma signal.

The revised analysis appears on lines 169-186 and Fig. 4 d-i.

Results & Methods

- My main concern with this paper is that most of the presented results (including the ‘travelling wave’ and ‘parallel processing’ analyses) do not rely on the high gamma power series, but on a statistical surrogate. Crucially, that surrogate is obtained by comparison to an arbitrary baseline period, defined with respect to saccade onset. Critically, this data (saccade-onset aligned and baseline-normalized) is used for all following analysis. This point is mentioned in the methods but seems to be forgotten when interpreting the results in the ‘travelling wave’ and ‘parallel processing’ sections.

o The authors explain that “In each frequency range, the signal modulation of the perisaccadic period was measured as t-statistics of each time point by comparison to baseline activity, which were median time points far after or before the saccade (1200 to 900 ms before saccade and 900 to 1200 ms after saccade)”. They indeed “obtained a single time series of the high-gamma signals as the average of t-statistics across 100–160 Hz”

o They justified this choice appealing to 1/f power-frequency relationship: “We analyzed this t-value time series data instead of the physical power of the raw ECoG signal because the high-frequency signal has small power in nature, but it still contains meaningful information in terms of the signal to noise ratio; thus, the difference in net signal power in each frequency range can be canceled out by computing the t-statistics, taking into account the signal to noise ratio in each frequency range”.

o This justification is inadequate as many forms of baseline normalization applied on power data per frequency (sub)band can solve this issue (z-score, dB).

o I do not understand why after going through the computational effort to run these statistics the authors used the resulting t-values but completely ignored the p-values from the test. Usually the ‘statistical approach’ is used to make use of only those time points and/or those channels that satisfy some threshold significance level (based on the p-values).

- In any case, the most problematic effect on some of their results comes from the fact that the used baseline is arbitrarily defined (between +/-900 and +/-1200 ms) with respect to saccade onset.

- Notice that saccades happen approximately every 200 ms in average; therefore, such distant baseline may not be ‘silent’ (as likely there are other saccades in the middle).

We appreciate the potential risk associated with the use of *t*-values with an arbitrary baseline as the surrogate signal, and that providing alternative approaches that are immune to such risks is a valuable approach. In the revised analysis, we calculated the z-score for each frequency band, and indeed the results were consistent with those reported in the original draft.

The changes relating to this point appear especially on Fig.2, Fig.8 and Supplementary Fig. 1

- It’s also problematic that the authors ran a two-tailed t-test to obtain this surrogate signal (obtaining both positive and negative t-values) and then excluded negative t-values for further analyses without offering any justification.

o Given the extensive evidence on the role of neural suppression in oculomotor behavior (REFS!), the decision to exclude negative t-values from their analyses of neural dynamics (at least for those shown in Figures 5 and 6) requires explanation.

o If the authors decided to exclude negative t-values (already a controversial choice), they could have used in principle a one-tail t-test to obtain the statistical surrogate.

o In the end, those calculated negative t-values are only displayed in the ‘statistical’ spectrograms shown in Figures 2 and 3 (which make them look similar to common power spectrograms).

o Correspondingly, their latency analyses seem to be identifying only positive peaks and ignoring suppression peaks.

o I would strongly recommend that instead of using the specified arbitrary period as baseline the authors compare perisaccadic time points in the actual data with a surrogate distribution obtained from randomly sampling the HGP time series. This procedure can be performed in the context of a permutation test and represents a robust approach that does not involve the selection of a specific and arbitrary baseline period. Alternatively, the authors could z-score the whole time-series corresponding to the recorded session after artifact rejection.

We added analysis of the negative components where appropriate and found interesting phenomena in addition to our original analysis, i.e., a consistent time difference between suppression and excitation along the ventral stream, triple dissociation across fixation/saccade onset, and excitation/suppression components in the MT and PPC.

In Figures 5 and 6(7 and 8 in the revised manuscript), our primary interest was to see to what extent the cortical activity evoked by a saccade remained before the next saccade was likely to occur. The inclusion of the negative component may underestimate the ongoing activation in other electrodes. In fact, for the electrodes whose activity returned to baseline, the high-gamma signal was further decreased below zero (z -score) (while this time course was different from the expected timing of saccade suppression). However, this observation does not alter our conclusion, i.e., the timing when the sum of positive activation across the entire brain returned to baseline corresponded to the timing of the most frequent inter-saccade interval in Figure 7b. Additionally, the signal transmission pattern did not change according to subsequent fixation duration, and thus the activation pattern at the moment of saccade onset was drastically different according to prior fixation duration (Fig. 8a). We included these arguments in the Methods section(line 507-512) and expanded them with Supplementary Figure 8.

- Signal Flow under natural visual behavior

- o The paper lacks a description of how latencies represented in Figs. 2 d,e were calculated. Was the peak defined by the local maximum? Were peak magnitudes and latencies detected per saccade and then averaged per site, or identified in the average over saccades per site? Please add this description to the methods section.

The latency and magnitude of the positive and negative peaks were obtained as local peaks after saccade onset. These were identified as averages over saccades per site. We were unable to obtain them in a trial-by-trial manner as the saccade-evoked gamma signal was subtle in each trial. We described this explicitly in the Methods section.

This change appears in the Methods section(line 436-439).

- Distinct patterns of motor- and visual-related signals

- o Several previous studies have functionally described single neurons as visual, motor or visuomotor in the context of oculomotor behavior. In the same line, the authors aim to quantify the contribution of visual and motor signals to the recorded LFP. For that purpose, the authors try to borrow a strategy from one previous single-cell study in which visual stimulus location was controlled in the context of two structured paradigms (Barash et al. 1991, using a double-saccade and back-saccade paradigm). This strategy consists in the distinction between motor and visual signals according to the comparative response magnitude to ipsi- and contralateral saccades (in relation to the brain hemisphere being recorded). In that single cell study, Barash et al. not only precisely control the location of the visual target, but also knows the receptive field and directional tuning of the single neurons they are studying in the lateral intra-parietal area (LIP).

- o In the present study by Kaneko et al. there is not a clear description of how visual stimuli location was controlled and it's unclear to me how their argument works in the absence of that information. The scheme in Figure 3a could be misleading by representing only one of the many possible cases. In a free visual exploration paradigm, depending on its starting point and its magnitude, and ipsilateral or contralateral saccade can bring certain part of the scene into or outside the (recorded) hemisphere receptive field.

- o Moreover, the authors' arguments are hard to follow and problematic especially with respect

to the afferent signal. For example, the authors assert that “afferent information should result in larger neural activity for ipsilateral saccades than for contralateral saccades”. Again, this does not hold true in their paradigm.

o Moreover, Kaneko’s argument does not take into account that different areas of the visual pathway are known to have differently sized ‘receptive fields’ (retinotopic selectivity).

o These considerations could explain why the authors did not find the expected afferent or efferent patterns in some areas of interest, and demands more careful interpretation of the data.

o Instead of relying on the conceptual scheme shown in Fig3a (appropriate only for more controlled paradigms at another level of description), the authors could compare activity patterns elicited by 2 possible LFP data alignments: saccade onset and fixation onset. This is the preferred approach in many published studies at the meso and macroscopic level.

As in the reply to the previous comments, we cannot deny that the interpretation of afferent processing only from ipsilaterally biased activity was not sufficiently convincing in the original draft. We appreciate the reviewer for providing an alternative approach that may reveal distinct profiles across dorsal regions showing a rapid response to saccades. In fact, we found an interesting dissociation across the MT and PPC areas. Although its function and precise computational meaning were unclear, we believe that this section now provides an important feature that differentiates MT and PPC activity during active visual behavior.

The changes appear on lines 112-153, 280-304, and Fig.4, 5 and Supplementary Fig.2.

• Spatiotemporal dynamics:

o According to the authors description, high-gamma spatiotemporal dynamics differed between the free viewing and the passive condition. The latter, only shown in S1F, exhibits the classically described pattern from occipital cortex to the dorsal and ventral streams. How do the authors precisely explain this difference?

We believe this may be derived from the limitation of the analysis approach we took in the present study. Our methods could identify the most prominent pattern of signal trajectory, and weaker trajectories might be masked by the most dominant route. During active visual exploration, the signal from the anterior to posterior direction is quite large in the PPC and dorsal regions. The neural signal may be transmitted from the early visual area toward the dorsal region, as suggested by the classic theory, but this was relatively less prominent and could not be shown in our analysis. We added this argument to the revised manuscript (lines 348-354).

Discussion

• After using the ‘traveling wave’ terminology repeatedly, the authors fail to include any reference to seminal work in the field. A growing number of papers has described the spatiotemporal dynamics of travelling-waves in humans and non-human primates. The literature on the functional role of travelling-waves elicited by visual stimulation is especially relevant in this case (REFS).

As in the reply to the previous comments, we added analysis of the travelling wave in phase gradient for the lower frequency oscillations. Indeed, we found that pattern of high-gamma signal trajectory is similar to that of travelling wave propagation. Thank you for the suggestion.

Figures

• Figure 2:

o The low frequency cut (y axis) in the statistical spectrograms shown in panel a, seems to be 50 Hz. Please add a label to the low limit, so it's clear activity below 50Hz is not shown. Fig2
We revised the figure so that the spectrogram we show is above 50 Hz.

o Two panels are labelled as (d). Fig2
We revised the organization of Figure 2.

• Figure 3 (This is Fig 4 in the revision)

o It would be beneficial to increase the size of the spectrograms in panel b. The spectrograms for ipsilateral and contralateral saccades are difficult to see and seem to be important for the distinction between afferent and efferent signals made by the authors. Fig3

□ Particularly, I'm intrigued by the apparent acute transition seen at time 0, even for 'background activity'. This is particularly obvious for MT panels (ipsilateral and contralateral saccades). Where each side of the spectrograms normalized to their one sided baseline? Fig3
We appreciate that the reviewer found the acute transition of the ipsilateral dominance of the MT interesting. However, we substantially revised this section to focus only on contralateral dominance, and not so much on ipsilateral dominance as the interpretation of ipsilateral bias is controversial (this was moved to the Discussion section as speculation). Our interpretation why the MT shows such a transition is as follows. First, the MT electrode may indeed respond more when the stimulus is present on the contralateral side of the visual field. Second, our stimulus was a $10 \times 10^\circ$ movie shown in front of our subject animal, so that the stimulus was sometimes inside/outside the receptive field population of the electrodes. Third, eye position was not controlled, so that gaze position on saccade onset was random across the movies; however, ipsilateral saccades, *on average*, tended to bring the stimulus inside the receptive field population of the MT electrode, and *vice versa* for contralateral saccades.

This argument appears on lines 284-292.

□ The baseline comparison period is included in these plots (-1200 to 900 ms and 900 to 1200 ms) and should correspond to zero, but that does not seem to be the case in the shown color scale. Fig3

As suggested by the reviewer, we now use z-scored signals instead of *t*-values with an arbitrary period as baseline, and the current analysis is immune from such biased activity long before or after saccade.

• Figure 4: (This is Fig 6 in the revision)

o The results shown in Figure 4 are definitely interesting, but the corresponding methods are not described. How was the center of gravity of the high gamma signal calculated? Please describe in the Methods section.

We substantially revised the corresponding paragraph in the manuscript. We added a simple formula by which we computed the center of gravity.

This change appears in the Methods section, lines 472-486

o Please show individual subject figures as Fig4B in the supplementary Information. This is important to verify that the general/whole trajectory dynamics are preserved at the single subject level.

We included the individual data in Supplementary Figure 3. As electrode density was sparse, we simply show the magnitude of activity at each electrode in each time bin. We believe that

the pattern observed as an across-subject average was consistent with the single subject level.

o As mentioned before, the data patterns observed in Figure 4 demand a phase-amplitude coupling analysis to test the hypothesis that high gamma power is coupled to the phase of theta oscillations (5 Hz). Further analysis would be required to make the point that those theta oscillations are indeed ‘travelling waves’ but given the extensive electrode coverage the authors are in the privileged position of performing those analyses.

We have added phase-amplitude coupling analysis. We indeed found that the high-gamma signal amplitude was coupled most with the phase of theta oscillations (~8 Hz in our study). We also found a spatiotemporal gradient of this theta phase on the cortical surface and indicated the direction of wave propagation. Interestingly, the directionality of the theta wave was similar to the trajectory of the high-gamma signal around saccade. These observations suggest that the high-gamma signal transmitted across the cortical surface was accompanied by the travelling wave of theta oscillations.

The new analysis appears in Fig. 6.

• Figure 5:

o A1 dynamics are shown as control. Including a site out of the visual pathway adds value to the results, but this control is surprisingly not mentioned or described in the main text. Do other sites far from the visual pathway trajectory look similar (flat)?

A certain level of saccade-evoked high-gamma modulation exists, but most of them were quite flat, in comparison to the visual areas, as in the example shown in the original text. We included this in Supplemental Figure 7.

• Figure 6:

o Were all recording sites included in the analysis corresponding to Figure 6a? Please provide a rationale for including sites out of the visual pathway.

We simply did not have strong motivation to include electrodes with weak saccade modulation as this will not change the pattern of whole brain dynamics. The same analysis dedicated only to the visual pathway confirmed that the pattern was similar to that shown in the original analysis. We added this result to Supplementary Figure 9.

REVIEWERS' COMMENTS:

Reviewer #1 (Remarks to the Author):

In this revision, the authors have addressed my comments raised in the first round of review. The additions of analysis on z-scored activity measures improve the manuscript and interpretability of the results (as noted to be necessary by the other reviewers).

Several minor comments should be addressed in the final version of the manuscript:

- As the authors note, the z-scores at the trial-averaged level are small, and this makes the HGP signal subtle to interpret. One potential cause for the smaller z-score values is signal variability from event to event. While outside the scope of results in the present manuscript, the authors could discuss opportunities for analyzing these recordings at the single-trial level in future work.

- The authors' updated discussion on LGN input to MT can still be more precise. The authors note that MT can be considered an additional primary visual area because it has "direct input" from LGN (cf. response to comment on line 239 in the original manuscript). While it is true that MT does receive direct input from LGN, other visual areas (e.g. V2 and V4) receive direct LGN input, as well; however, the LGN input to V2 and V4 is relatively weak (see Sincich et al., Nature Neuroscience 7, 2004 for a discussion on this). With this in mind, the authors should consider to clarify this point by adding an appropriate qualifier, such as "strong, direct input from LGN", so that this statement is more accurate.

Reviewer 3 shared many of my concerns with the manuscript. The first of these concerns was the analysis based on t-statistics, which I believe the authors have addressed in their revision. The second major point was to include the analysis of a traveling wave in the lower frequency range, which the authors have also addressed well.

-The third major point concerns afferent and efferent inputs during peri-saccadic activity. On this point, the authors have substituted the original analysis (which was correctly flagged by the reviewer as problematic) with a more general statement on high-gamma modulations around the saccade.

-As Reviewer 3 is clearly an expert on saccades, I am somewhat uncomfortable speaking fully for them on this third point. At the same time, however, I believe the authors' changes do address all concerns, provided that they systematically avoid discussion of afferent and efferent inputs during saccades. For example, with the evidence presented here, I think the authors can state there is a clear difference between MT and PPC regions, but cannot make conclusions about the underlying mechanism.

Reviewer #2 (Remarks to the Author):

The authors have made substantial revisions to their paper and I believe the major concerns are addressed in the current revision. Previously I had confusion about the lack of clarity in interpretation of "afferent vs efferent" signals, which now have been removed and replaced with more specific claims about retinotopic characteristics of the dorsal and ventral pathway areas. The authors have also done a nice job with implementing follow up analysis of low frequency traveling waves and showing the co-occurrence of high gamma activity with the ongoing theta oscillations. We believe the corresponding analysis in figure 6(d-f) increases the impact of the paper and will be beneficial for ongoing research in the field of traveling waves. The current text uses more consistent terminology and avoids using undefined synonyms that we pointed out in the first round of review process. Lastly, the paper is also improved by the authors including a replication of figure 8b for each of 4 animals. As we expected, focusing on individual animals and selecting shorter post-saccade intervals for the figures 6.B, clarifies regional differences of gamma activity that were difficult to distinguish in previous revision.

Based on reading the manuscript, I have two suggestions that the authors might want to include in the final revision of the paper:

* In their letter the authors responded to my point regarding the apparent relationship between fixation duration and the power of subsequent peaks in Figure 6A. This same explanation would be useful to mention in the text of the results or the corresponding discussion of Figure 6. I think this might help eliminate confusion on the part of the reader.

* A major significant result of the paper relates to the different dynamics of brain activity during active vision vs passive vision. I think it would be useful to mention this in the main text rather than just the supplement. Merging supplementary Fig4.d with Fig 6.a in a new figure would make this point.

We would like to thank the editor and reviewers for their constructive comments, which helped us to improve our manuscript. Below, we address all of their comments in a point-by-point manner and discuss the subsequent revisions.

The original comments from the reviewers are shown in black font, and our replies are shown in blue font.

Reviewer #1:

· As the authors note, the z-scores at the trial-averaged level are small, and this makes the HGP signal subtle to interpret. One potential cause for the smaller z-score values is signal variability from event to event. While outside the scope of results in the present manuscript, the authors could discuss opportunities for analyzing these recordings at the single-trial level in future work.

Thank you for the suggestion. We agree that single trial analysis would be a powerful approach to explain variability in signal strength, especially for the naturalistic behavioral paradigm. We added the following statement in the discussion.

*Schall et al.⁴⁷ proposed a beautiful schema showing an anatomical connectivity gradient in the anterior bank of the arcuate sulcus that coexists with a functional gradient in saccade amplitude at the frontal eye field. The visual system is designed to work under natural behavior, so there must be a reasonable link across different types of saccade (amplitude, interval, and target visual features), its underlying neural architecture, and the visual environment. **Our current analysis focused on neural dynamics of the trials' averaged signals. Analysis at a single-trial level may reveal links between behavior, neural architecture, and the environment in future work.***

· The authors' updated discussion on LGN input to MT can still be more precise. The authors note that MT can be considered an additional primary visual area because it has "direct input" from LGN (cf. response to comment on line 239 in the original manuscript). While it is true that MT does receive direct input from LGN, other visual areas (e.g. V2 and V4) receive direct LGN input, as well; however, the LGN input to V2 and V4 is relatively weak (see Sincich et al., Nature Neuroscience 7, 2004 for a discussion on this). With this in mind, the authors should consider to clarify this point by adding an appropriate qualifier, such as "strong, direct input from LGN", so that this statement is more accurate.

Thank you for your comment. We have revised as suggested, and think it will improve the accuracy of our description.

· As Reviewer 3 is clearly an expert on saccades, I am somewhat uncomfortable speaking fully for them on this third point. At the same time, however, I believe the authors' changes do address all concerns, provided that they systematically avoid discussion of afferent and efferent inputs during saccades. For example, with the evidence presented here, I think the authors can state there is a clear difference between MT and PPC regions, but cannot make conclusions about the underlying mechanism.

Reviewer #2:

· In their letter the authors responded to my point regarding the apparent relationship between fixation duration and the power of subsequent peaks in Figure 6A. This same explanation would be useful to mention in the text of the results or the corresponding discussion of Figure 6. I think this might help eliminate confusion on the part of the reader.

Thank you for the suggestion. We agree that this argument is important to avoid confusion in readers who are interested in this trend. We added our previous reply in the result section as below.

In figure 9a, there appears to be a relationship between fixation duration and the power of subsequent peaks. We are not sure for the precise reason why the power of subsequent peaks differed across fixation duration. The signal pattern after fixation initiation (0–200 ms; 0–130 ms for the 150 ms fixation condition) was quite similar regardless of how long the subsequent fixation was maintained, however, the activity profiles just before the next saccade differed drastically according to the prior fixation duration. This difference might explain the difference in the power of the subsequent peak.

· A major significant result of the paper relates to the different dynamics of brain activity during active vision vs passive vision. I think it would be useful to mention this in the main text rather than just the supplement. Merging supplementary Fig4.d with Fig 6.a in a new figure would make this point.

We combined Supplementary Figure 4d into Figure 6 as suggested. As the size of figure 6 became larger, we divided them into two figures to preserve flexibility of the page layout.